# Egr2 drives the differentiation of Ly6C[hi] monocytes into fibrosis-promoting macrophages in metabolic dysfunction-associated steatohepatitis in mice
Ayaka Iwata[1], Juri Maruyama[1], Shibata Natsuki[1], Akira Nishiyama [2], Tomohiko Tamura [2,3], Minoru Tanaka [4], Shigeyuki Shichino [5], Takao Seki [6], Toshihiko Komai[7], Tomohisa Okamura [7], Keishi Fujio [7], Masato Tanaka [1,8] ✉ & Kenichi Asano [1,8] ✉

Metabolic dysfunction-associated steatohepatitis (MASH), previously called non-alcoholic steatohepatitis (NASH), is a growing concern worldwide, with liver fibrosis being a critical determinant of its prognosis. Monocyte-derived macrophages have been implicated in MASH-associated liver fibrosis, yet their precise roles and the underlying differentiation mechanisms remain elusive. In this study, we unveil a key orchestrator of this process: long chain saturated fatty acid-Egr2 pathway. Our findings identify the transcription factor Egr2 as the driving force behind monocyte differentiation into hepatic lipid-associated macrophages (hLAMs) within MASH liver. Notably, *Egr2*-deficiency reroutes monocyte differentiation towards a macrophage subset resembling resident Kupffer cells, hampering hLAM formation. This shift has a profound impact, suppressing the transition from benign steatosis to liver fibrosis, demonstrating the critical pro-fibrotic role played by hLAMs in MASH pathogenesis. Long-chain saturated fatty acids that accumulate in MASH liver emerge as potent inducers of Egr2 expression in macrophages, a process counteracted by unsaturated fatty acids. Furthermore, oral oleic acid administration effectively reduces hLAMs in MASH mice. In conclusion, our work not only elucidates the intricate interplay between saturated fatty acids, Egr2, and monocyte-derived macrophages but also highlights the therapeutic promise of targeting the saturated fatty acid-Egr2 axis in monocytes for MASH management.

Non-alcoholic fatty liver disease, recently renamed as metabolic dysfunction-associated steatotic liver disease (MASLD)[1], is a spectrum of disorders characterized by excessive lipid accumulation in hepatocytes. MASLD progresses from fatty liver (steatosis) to metabolic dysfunction-associated steatohepatitis (MASH), and in more severe cases, to liver fibrosis[2]. Although the prognosis of patients with MASLD is strongly associated with the degree of fibrosis[3,4], the mechanism underlying the transition from simple steatosis to fibrosis remains unclear, making the development of anti-MASH therapy a huge hurdle. Hepatic stellate cells (HSCs) and myofibroblasts that differentiate from HSCs are considered to play a predominant role in the progression to liver fibrosis[5]. Recently, not only such stromal cells but also immune cells, macrophages in particular, are found to contribute to the pathogenesis of liver fibrosis by regulating the activation states of HSCs[6]. In the steady state, Kupffer cells (KCs), a major

[1]Laboratory of Immune Regulation, School of Life Sciences, Tokyo University of Pharmacy and Life Sciences, Tokyo 192-0392, Japan. [2]Department of Immunology, Yokohama City University Graduate School of Medicine, Kanagawa 236-0004, Japan. [3]Advanced Medical Research Center, Yokohama City University, Kanagawa 236-0004, Japan. [4]Department of Regenerative Medicine, Research Institute National Center for Global Health and Medicine, Tokyo 162-8655, Japan. [5]Division of Molecular Regulation of Inflammatory and Immune Diseases, Research Institute for Biomedical Sciences, Tokyo University of Science, Chiba 278-0022, Japan. [6]Department of Biochemistry, Toho University School of Medicine, Tokyo 143-8540, Japan. [7]Department of Allergy and Rheumatology, Graduate School of Medicine, The University of Tokyo, Tokyo 113-0033, Japan. [8]These authors contributed equally: Masato Tanaka, Kenichi Asano. ✉e-mail: mtanaka@toyaku.ac.jp; asanok@toyaku.ac.jp

macrophage subset in liver, reside along the length of liver sinusoid where they contribute to the clearance of toxic materials from bloodstream. One of the core features of MASH is the massive accumulation of blood-borne monocytes[7] that further differentiate into monocyte-derived macrophages, replacing the resident KCs[8–11]. Known as scar-associated macrophages (SAMs)[12,13], these recruited monocyte-derived cells best resemble lipid-associated macrophages (LAMs) that aggregate to surround dead adipocytes in obese adipose tissue[14], and are therefore named hepatic LAMs (hLAMs)[8]. There are reports that SAMs crosstalk with various non-parenchymal cells, thereby contributing to the progression of MASH to liver fibrosis in both mouse and human[13,15]. However, the impact of monocyte-derived macrophages on the progression to liver fibrosis is still poorly understood because the inhibition of Ccr2-mediated monocyte recruitment can also aggravate the progression to liver fibrosis in mice fed a high-fat diet[11]. There is an urgent need to identify the regulatory mechanisms controlling monocyte-derived macrophage behavior to understand their function in promoting or suppressing the progression of MASH to liver fibrosis.

To understand the mechanisms driving the differentiation of monocyte-derived macrophages, we identified Egr2, a transcription factor that is enhanced in liver monocytes and macrophages in mice with MASH. Liver fibrosis was ameliorated in mice deficient in *Egr2* in immune cells of myeloid lineage. Profibrotic macrophage signatures[16] including *Spp1*, *Fabp5*, and *Cd63*, but not *Trem2*, were repressed in *Egr2*-deficient liver macrophages. Single-cell RNA-sequencing (scRNA-seq) analysis showed that the proportion of hLAMs in the monocyte-derived macrophage pool was reduced, and the differentiation of monocytes was biased toward another subset of macrophages expressing high levels of KC signatures that are distinct from either hLAMs or resident KCs owing to the absence of *Egr2*. In this study to identify niche-specific environmental factors regulating *Egr2* expression in liver infiltrating monocytes, we found that long-chain fatty acids upregulate Egr2 in macrophages. Moreover, oral administration of long-chain unsaturated fatty acid decreased the proportion of hLAMs liver macrophages in vivo.

Through this study, we have uncovered the role of Egr2 in the generation of profibrotic macrophages from monocytes. This knowledge may be applied to the development of Egr2-targeted therapy for preventing fibrosis in patients with MASLD.

## Results

### *Egr2* expression is progressively enhanced in liver macrophages during the development of MASH

To analyze the relationship between the development of liver fibrosis and monocyte-derived macrophages, we used a mouse model of choline-deficient amino acid-defined high-fat diet (CDA-HFD)-induced MASH[17] that rapidly progresses to liver fibrosis. First, we analyzed the progression of MASLD in CDA-HFD-fed mice. Microscopically, feeding of CDA-HFD promoted hepatocellular steatosis in 7 weeks (Fig. 1a, left). Compared to HFD which takes 6 months for liver fibrosis to develop[18], liver fibrosis was detectable at week 7 and further progressed within the next 5 weeks by feeding CDA-HFD (Fig. 1a-c). Next, we analyzed the composition of immune cells in CDA-HFD-induced MASH liver. Massive infiltration of Lin (B220, CD90.2, NK1.1, and SiglecF)$^-$CD11b$^+$Ly6G$^-$Ly6C$^{hi}$F4/80$^-$ monocytes into liver was observed in 4 weeks (Fig. 1d, middle and Supplementary Fig. 1a). Lin$^-$CD11b$^+$Ly6C$^{lo}$F4/80$^+$Tim4$^+$ resident KCs were rapidly replaced by Lin$^-$CD11b$^+$Ly6C$^{lo}$F4/80$^+$Tim4$^-$ monocyte-derived macrophages in 4 weeks and were barely detectable as early as week 7 (Fig. 1d, middle and right). Immunohistochemistry also demonstrated that Tim4 expression was reduced in macrophages in MASH liver (Fig. 1e). Localization of macrophages was drastically changed in MASH liver, that is, aggregates of F4/80$^+$ macrophages are forming crown-like structures[11,19] that may be surrounding dead hepatocytes in MASH liver (Fig. 1e).

Given that most resident KCs were replaced by Tim4$^-$ monocyte-derived macrophages before the progression to liver fibrosis, we speculated

that the phenotype of monocyte-derived macrophages at 7 weeks might be associated with the transition from steatosis to fibrosis. To understand the molecular mechanisms regulating the phenotype of monocyte-derived macrophages in MASH liver, we performed bulk RNA-seq analysis of liver Ly6C$^{lo}$F4/80$^+$ macrophages in normal diet (ND)-fed mice and CDA-HFD-fed wild type (WT) mice. Gene expression of liver macrophages was dramatically changed in mice fed CDA-HFD (Fig. 1f); this was characterized by the upregulation of several markers for classic SAMs such as *Trem2*, *Cd9*, and *Spp1*, and the downregulation of KC markers such as *Clec4f* and *Timd4*[13] (Fig. 1g). We then sought transcription factors that were enhanced in liver macrophages of CDA-HFD-fed mice when compared with ND-fed mice (Fig. 1h). Among such transcription factors, we focused on Egr2 because it was most strongly upregulated in monocytes after infiltration into liver at week 7 (Fig. 1i) and progressively enhanced in macrophages during the fibrotic stage of MASLD (Supplementary Fig. 1b). At week 4 when liver macrophages included both Tim4$^+$ resident KCs and Tim4$^-$ monocyte-derived macrophages, *Egr2* mRNA was more highly expressed in Tim4$^-$ monocyte-derived macrophages than in Tim4$^+$ resident KCs (Supplementary Fig. 1c), suggesting that Egr2 may play a role in the differentiation of Tim4$^-$ monocyte-derived macrophages in MASLD. Overall, these findings indicate that Egr2 expression in monocyte-derived macrophages is positively correlated with the progression of MASH to liver fibrosis.

### Liver fibrosis is ameliorated in mouse lacking *Egr2*

To determine the role of Egr2 in monocyte-derived macrophage differentiation in MASH, we crossed *Lyz2*-Cre mice with *Egr2*-flox mice[20] to generate a strain (hereinafter called *Lyz2*$^{Cre/+}$.*Egr2*$^{fl/fl}$ mice) in which monocytes and macrophages, but not lymphoid cells, lack *Egr2*[21] (Fig. 2a and Supplementary Fig. 2a). First, we compared the composition of immune cells in the liver of MASH-induced *Lyz2*$^{Cre/+}$.*Egr2*$^{fl/fl}$ mice and MASH-induced *Egr2*$^{fl/fl}$ littermates by flow cytometry. The percentages of Ly6C$^{hi}$F4/80$^-$ monocytes and Ly6C$^{lo}$F4/80$^+$Tim4$^-$ monocyte-derived macrophages in liver were similar between the two genotypes of mice at both 7 and 12 weeks (Fig. 2b and Supplementary Fig. 2b). These results indicate that Egr2 is not required for the recruitment of monocytes. Next, we compared the progression of MASLD between *Lyz2*$^{Cre/+}$.*Egr2*$^{fl/fl}$ mice and *Egr2*$^{fl/fl}$ littermates. The absence of *Egr2* did not change the degree of inflammation or steatosis (Fig. 2c–f and Supplementary Fig. 2c, d). On the other hand, a marked difference in the degree of fibrosis was evident at week 12 (Fig. 2g). As shown in Fig. 2h, the amount of collagen was decreased in the liver of CDA-HFD-fed *Lyz2*$^{Cre/+}$.*Egr2*$^{fl/fl}$ mice compared with WT littermates. Amelioration of fibrosis was validated also on the basis of fibrosis-associated gene expression levels in tissue (Fig. 2i). Collectively, these findings indicate that the Egr2 expression by monocytes and macrophages is responsible for the development of liver fibrosis in MASH.

### Monocyte-derived macrophages fail to acquire profibrotic phenotype in the absence of *Egr2*

To assess the differences between monocyte-derived macrophages in MASH-induced *Egr2*$^{fl/fl}$ mice and *Lyz2*$^{Cre/+}$.*Egr2*$^{fl/fl}$ mice, we performed bulk RNA-seq analysis of CD11b$^+$ Ly6G$^-$Ly6C$^{lo}$F4/80$^+$ monocyte-derived macrophages sorted from the liver of *Lyz2*$^{Cre/+}$.*Egr2*$^{fl/fl}$ mice and *Egr2*$^{fl/fl}$ littermates. Unbiased clustering confirmed that the biological replicates from each group adopted a distinct transcriptional signature (Fig. 3a). Differential gene expression analysis revealed that 520 genes were differentially expressed by at least twofold (123 and 397 genes downregulated and upregulated in *Lyz2*$^{Cre/+}$.*Egr2*$^{fl/fl}$ mice compared with *Egr2*$^{fl/fl}$ mice, respectively, Fig. 3b, c). We confirmed that known downstream targets of Egr2 (*Itgax* and *Siglecf*) were indeed repressed in *Lyz2*$^{Cre/+}$.*Egr2*$^{fl/fl}$ mice (Supplementary Fig. 3a). Gene set enrichment analysis (GSEA) showed that the global gene expression of WT monocyte-derived macrophages was related to classic SAM signatures (Fig. 3d, left and 3e, top). On the other hand, the gene expression of *Egr2*-deficient monocyte-derived macrophages was enriched for KC

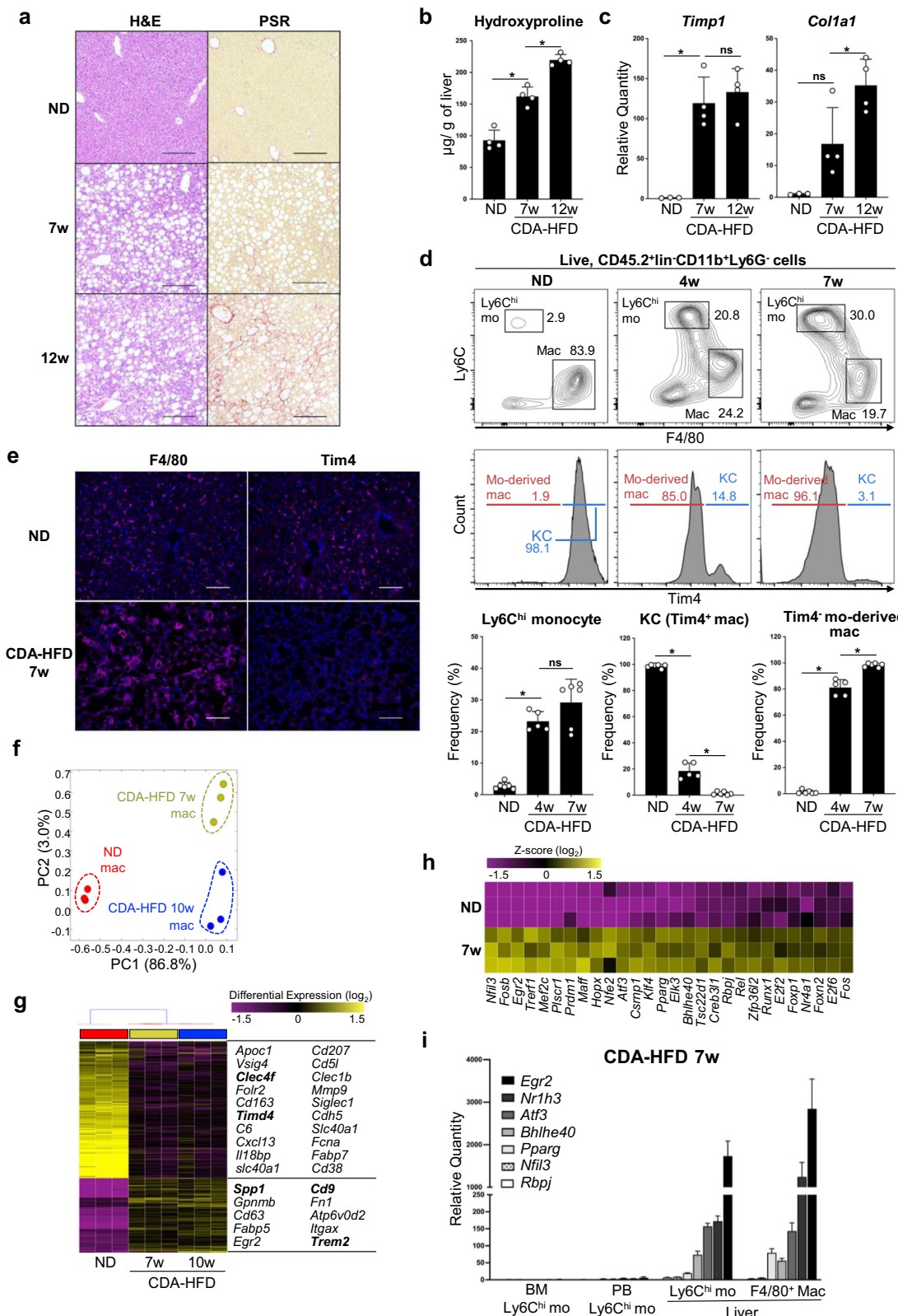

signatures (Fig. 3d, right and 3e, bottom). Of note was that the expression levels of some key SAM signatures, such as *Trem2*, were similar between the two genotypes of macrophages (Fig. 3e, top). As the upregulation of KC markers in *Egr2*-deficient Ly6C$^{lo}$F4/80$^+$ macrophages cannot be explained by the increase of Tim4$^+$ resident KCs at 7 weeks (Fig. 2b, also see scRNA-seq data discussed later in Fig. 4a), these findings suggest that

Egr2 is required for the full equipment of SAM phenotype by monocyte-derived macrophages, which may promote the transition from simple steatosis to fibrosis over the course of MASLD. Genes upregulated in *Egr2*-deficient macrophages were enriched for gene ontology (GO) terms associated with vasculature and blood vessel development, suggesting their commitment to tissue regeneration (Fig. 3f).

**Fig. 1 | Egr2 is upregulated in liver-infiltrating monocytes and monocyte-derived macrophages in MASH liver. a** Histopathological changes in the liver of mice fed CDA-HFD. H&E staining showed steatosis at 7 weeks (middle left) and 12 weeks (bottom left). Fibrosis was not detected in mice fed normal diet (ND, top right) or mice-fed CDA-HFD (middle right) for 7 weeks. Picrosirius red (PSR) staining showed apparent fibrosis at 12 weeks (bottom right). Representative images of two mice are shown. Scale bars, 200 μm, original magnification, ×10. **b** The amount of collagen in the liver was positively correlated with the progression of MASH. n = 4 mice/time point. Means ± SD are shown. *$p < 0.05$, one-way ANOVA. **c** Fibrosis-associated gene expression was measured by qPCR. n = 3–4 mice/time point. Mean quantities relative to ND-fed mice ± SD are shown. *$p < 0.05$, one-way ANOVA. **d** Percentages of liver monocytes, KCs, and monocyte-derived macrophages in CD45.2+lin⁻CD11b+Ly6G⁻ cells. Lin includes CD90.2, B220, NK1.1, and SiglecF. Representative FACS plots of five to six mice per time point are shown (top). Histogram plot showing the proportions of Tim4+ resident KCs and Tim4⁻ monocyte-

derived macrophages (mo-derived mac, middle and bottom). n = 5-6 mice/time point. Means ± SD are shown. *$p < 0.05$, one-way ANOVA. **b–d** Each symbol represents an individual animal. **e** Immunohistochemistry of ND and CDA-HFD-fed mouse liver. Representative images of 2 different mice are shown. Scale bars, 100 μm, original magnification, ×20. **f–h** RNA-seq analysis of liver macrophages. **f** Principal component analysis (PCA) was performed to visualize gene expression in liver macrophages from ND-fed mice (red) or CDA-HFD-fed mice (7 weeks, yellow or 10 weeks, blue). Each symbol represents an individual animal. **g** Hierarchical cluster analysis showing the relative expression of discriminative genes across the different time points. Each column represents an individual animal. Representative DEGs are listed on the right. **h** Heatmap showing the expression of transcription factors upregulated in liver macrophages from CDA-HFD-fed mice relative to that from ND-fed mice. Each row represents an individual animal. **i** Expression of transcription factors selected from Fig. 1h was measured by qPCR. Average quantities relative to BM Ly6Chi mo are shown with SEM. n = 3 mice/gene/cell type.

## Heterogeneity of monocyte-derived macrophages in MASH liver

Given that the gene expression of monocyte-derived macrophages was enriched for KC markers in *Lyz2*^Cre/+.*Egr2*^fl/fl mice (Fig. 3d, e), although resident Tim4+ KCs were almost completely replaced by Tim4⁻ monocyte-derived macrophages by week 7 (Fig. 2b), the differentiation of monocyte-derived macrophages might be altered by *Egr2*-deficiency in MASH. Hence, to gain an insight into the global effects of *Egr2* deletion on monocyte-derived macrophage differentiation in MASLD liver, we performed scRNA-seq analysis of liver innate immune cells. To this end, we sorted liver CD45+Lin (B220, CD90.2, NK1.1, and SiglecF)⁻CD11b+ cells of an *Egr2*^fl/fl mouse and a *Lyz2*^Cre/+.*Egr2*^fl/fl mouse fed CDA-HFD for 7 weeks and performed sequencing using 10x Chromium platform. Fifteen thousand three hundred and ninety-four cells passed quality control and were clustered using Uniform Manifold Approximation and Projection (UMAP) dimensionality reduction analysis[22] within Seurat v4 (Fig. 4a). Cell types were unbiasedly annotated by using SingleR[23] (Supplementary Fig. 4a) or manually annotated on the basis of DEGs (Supplementary Fig. 4b, c). scRNA-seq analysis showed the expansion of neutrophils (Clusters 5, 8, and 16), Ly6Chi monocytes (Cluster 4), dendritic cells (Clusters 1, 12, and 13), and monocyte-derived macrophages (Clusters 2 and 7) in MASH (Fig. 4a and Supplementary Fig. 4d). Resident KCs (Clusters 3 and 14) accounted for the largest population in the steady state in both *Egr2*^fl/fl and *Lyz2*^Cre/+.*Egr2*^fl/fl mice (Fig. 4a, top right, and Supplementary Fig. 4d). In MASH, on the other hand, KCs were rarely observed in both mouse genotypes (Fig. 4a and Supplementary Fig. 4d). Monocytes and monocyte-derived macrophage clusters that were distinct from resident KCs were dramatically expanded in both mouse genotypes in MASH (Fig. 4a and Supplementary Fig. 4d). These data clearly exclude the possibility that the upregulated expression of KC markers by *Egr2*-deficient macrophages in bulk RNA-seq analysis (Fig. 3d, e) may be ascribed to the expansion of KCs within the Ly6Clo F4/80+ macrophage cluster. The proportions of myeloid cells were similar between the two genotypes of mice in MASH (Fig. 4a and Supplementary Fig. 4d). To precisely analyze the phenotypes of monocytes and monocyte-derived macrophages in MASH, KCs, NK cells, neutrophils, and dendritic cells were excluded, and the remaining cells were re-clustered to yield 11 clusters of monocytes and Tim4⁻ monocyte-derived macrophages (Fig. 4b). Clusters 1, 4, and 7 represent Ly6Chi monocytes based on their high expression of *Ly6c2* and *Ccr2* (Fig. 4b–d). Clusters 5 and 0 represent Ly6Clo monocytes and Ly6Cint monocytes, respectively (Fig. 4b–d). We found that Tim4⁻ monocyte-derived macrophages were separated into clusters 3 and 6 in MASH (Fig. 4b–d). Cluster 3 was slightly expanded, whereas cluster 6 was apparently reduced in *Lyz2*^Cre/+.*Egr2*^fl/fl mice compared with *Egr2*^fl/fl mice (Fig. 4b, e).

To understand the relationship between Egr2 expression and the composition of monocyte-derived macrophages, we examined whether clusters 3 and 6 correspond to any one of the monocyte-derived macrophages reported previously. Remmerie's group separated liver monocyte-derived macrophages into hLAMs (cells that correspond to SAMs) and monocyte-derived KCs (moKCs)[8], although the functions of each subset

were not precisely defined. We calculated the expression of marker genes for hLAMs and moKCs by Tim4⁻ monocyte-derived macrophage clusters 3 and 6 generated by our scRNA-seq analysis by using AddModuleScore[24]. This analysis revealed that cluster 6 aligned closely with the gene expression pattern of hLAMs, which was characterized by the increased expression of *Spp1*, *Fabp5*, *Cd9*, and *Egr2* (Fig. 4d, f, left and Supplementary Fig. 4e). On the other hand, cluster 3 showed transcriptional similarities to moKCs that are characterized by increased moKC signatures such as *Il18bp* and *Clec4f* (Fig. 4d, f, right). Altogether, these results suggest an important role played by Egr2 in controlling the proportions of hLAMs and moKCs subpopulations in the monocyte-derived macrophage pool.

## Egr2 drives the differentiation of monocytes into hLAMs in MASH liver

On the basis of these findings, we speculated that Egr2 favored the differentiation of liver-infiltrating monocytes into hLAMs, whereas the differentiation pathway was shifted toward moKCs in the absence of *Egr2*. To infer the mechanism underlying the alteration of differentiation pathways of *Egr2*^fl/fl and *Lyz2*^Cre/+.*Egr2*^fl/fl monocyte-derived macrophages, we performed trajectory analysis by using RNA velocity (scVelo)[25,26]. This analysis suggested that Ly6Cint monocytes differentiated mainly into cluster 6 in *Egr2*^fl/fl mouse (Fig. 5a, top left). On the other hand, in *Lyz2*^Cre/+.*Egr2*^fl/fl mouse, the differentiation pathways from monocytes and cluster 6 were directed toward cluster 3 (Fig. 5a, top right), suggesting the role of Egr2 in driving the differentiation of monocytes into hLAMs and in inhibiting the differentiation of hLAMs into moKCs (Fig. 5a, bottom). To validate this computational prediction in vivo, BM monocytes from ND-fed CD45.2+ *Lyz2*^Cre/+.*Egr2*^fl/fl mouse or *Egr2*^fl/fl littermate were transferred into CDA-HFD-fed CD45.1+ recipients (Fig. 5b, top). The kinetics of the transferred monocytes in the recipient liver was monitored by flow cytometry. This analysis showed that both *Lyz2*^Cre/+.*Egr2*^fl/fl and *Egr2*^fl/fl monocytes differentiated into Ly6Clo F4/80+ monocyte-derived macrophages in 3 days (Fig. 5b, bottom). We found that *Egr2*^fl/fl monocyte-derived Ly6Clo F4/80+ macrophages expressed hLAM-associated genes such as *Spp1* and *Fabp5* in high levels compared with *Lyz2*^Cre/+.*Egr2*^fl/fl monocyte-derived ones (Fig. 5c). Collectively, these results validated the computer prediction that Egr2 is required for the full maturation of hLAMs.

## Egr2 promotes the differentiation of monocytes into lipid-rich proinflammatory macrophages

To analyze the characteristics of different clusters (clusters 3 and 6) of monocyte-derived macrophages, we sought surface markers that can discriminate the two subsets by flow cytometry. Among the DEGs (Supplementary Data 1) that were upregulated in cluster 6, we found that a combination of CD11c and Mincle was able to separate Tim4⁻CD11b+Ly6Clo F4/80+ monocyte-derived macrophages into CD11chiMinclehi and CD11cloMinclelo subfractions (Fig. 6a). The percentage of CD11chiMinclehi macrophages in Ly6ChiF4/80+ macrophages was positively correlated with the progression of MASH (Supplementary Fig. 5a). To

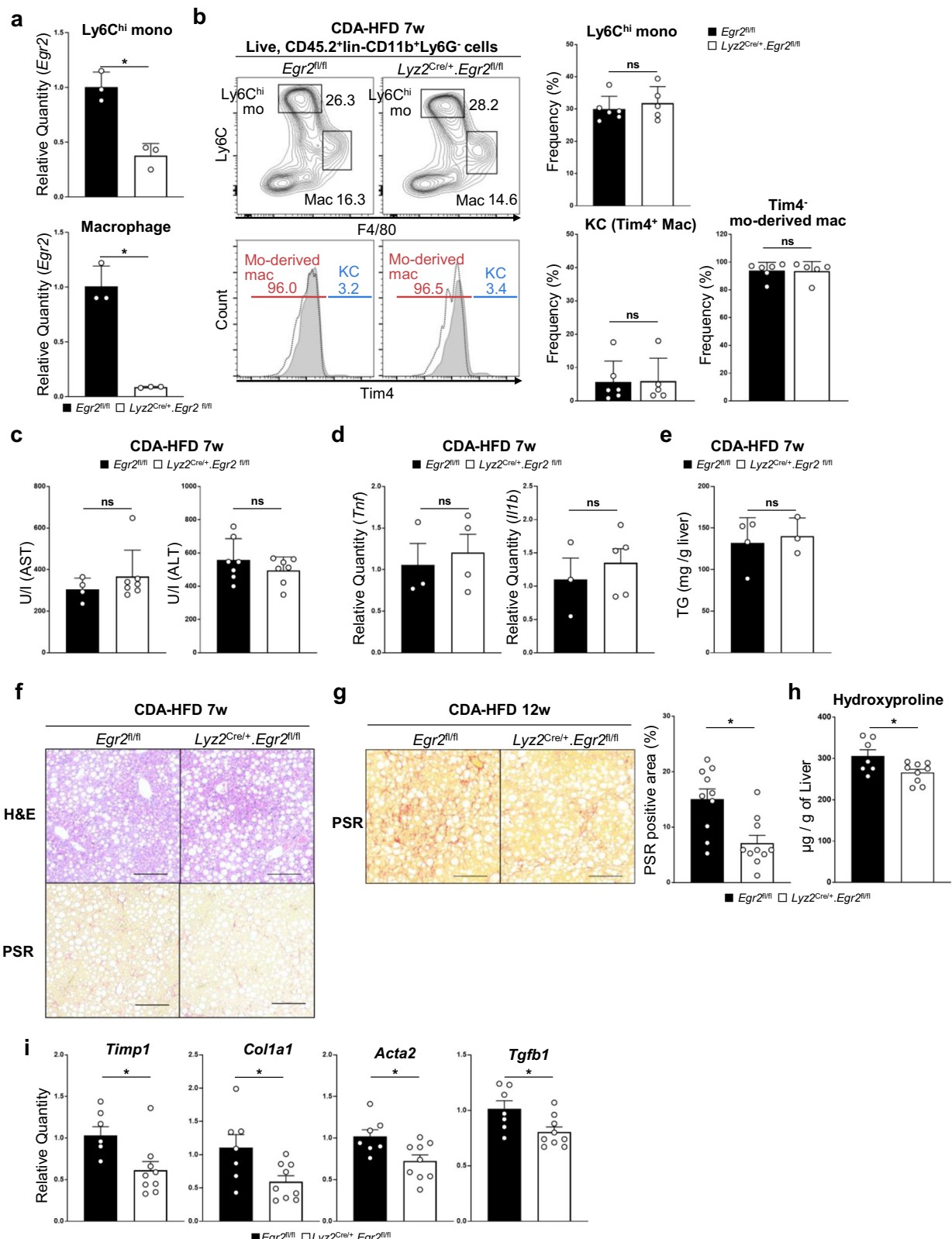

confirm whether CD11c$^{hi}$Mincle$^{hi}$ Ly6C$^{lo}$F4/80$^+$ cells and CD11c$^{lo}$Mincle$^{lo}$ Ly6C$^{lo}$F4/80$^+$ cells correspond to monocyte-derived macrophage clusters defined by scRNA-seq analysis, we sorted Tim4$^-$Ly6C$^{lo}$F4/ 80$^+$CD11c$^{hi}$Mincle$^{hi}$ and Tim4$^-$Ly6C$^{lo}$F4/80$^+$CD11c$^{lo}$Mincle$^{lo}$ cells, examined their gene expression by RNA-seq analysis, and compared the relative expression of signature genes discriminating clusters 3 and 6 defined by

scRNA-seq and bulk RNA-seq data. Bulk RNA-seq analysis of Tim4$^-$Ly6C$^{lo}$F4/80$^+$CD11c$^{hi}$Mincle$^{hi}$ cells yielded expression profiles concordant to those of cluster 6 defined by scRNA-seq analysis when marker genes were compared, whereas bulk RNA-seq analysis of Tim4$^-$Ly6C$^{lo}$F4/ 80$^+$CD11c$^{lo}$Mincle$^{lo}$ cells gave expression profiles that were aligned with those of cluster 3 (Fig. 6b). GSEA showed that CD11c$^{hi}$Mincle$^{hi}$

**Fig. 2 | Liver fibrosis is ameliorated in *Egr2*-deficient mice. a** *Egr2* mRNA expression in Ly6C$^{hi}$F4/80$^-$ monocytes (top) and Ly6C$^{lo}$F4/80$^+$ macrophages (bottom) in CDA-HFD-fed *Egr2*$^{fl/fl}$ mice (filled bar) and *Lyz2*$^{Cre/+}$.*Egr2*$^{fl/fl}$ (empty bar) mice. Average quantities relative to *Egr2*$^{fl/fl}$ mice are shown together with SD. n = 3 mice. *$p$ < 0.05, *t*-test. **b** Percentages of liver monocytes and macrophages in CD45.2$^+$lin$^-$CD11b$^+$Ly6G$^-$ cells from *Egr2*$^{fl/fl}$ (filled bar) and *Lyz2*$^{Cre/+}$.*Egr2*$^{fl/fl}$ (empty bar) mice fed CDA-HFD for 7 weeks. Lin includes CD90.2, B220, NK1.1, and SiglecF. Representative FACS plots of five to six mice are shown (left). Histogram plot showing the proportions of Tim4$^+$ resident KCs and Tim4$^-$ mo-derived mac in Ly6C$^{lo}$F4/80$^+$ cells (bottom left). Dashed line indicates Tim4 unstained control. Averages of 5–6 mice/genotype are shown with SD. ns not significant, *t*-test. **c** Serum AST and ALT concentration in *Egr2*$^{fl/fl}$ (filled bar) mice and *Lyz2*$^{Cre/+}$.*Egr2*$^{fl/fl}$ (empty bar) mice fed CDA-HFD for 7 weeks. n = 4–7 mice/genotype. Means ± SD are shown. ns not significant, *t*-test. **d** Whole-liver qPCR analysis of *Tnf* in *Egr2*$^{fl/fl}$ (filled bar) and *Lyz2*$^{Cre/+}$.*Egr2*$^{fl/fl}$ (empty bar) mice fed CDA-HFD for 7 weeks. n = 3–5 mice/genotype. Means ± SEM are shown. ns not significant, *t*-test. **e** Amount of triglyceride in liver. n = 3–4 mice/genotype. Means ± SD are shown. ns not significant, *t*-test. **f** Histopathological changes in the liver of *Egr2*$^{fl/fl}$ mice (left) and *Lyz2*$^{Cre/+}$.*Egr2*$^{fl/fl}$ mice (right) fed CDA-HFD for 7 weeks. Steatosis (top) was comparable between *Egr2*$^{fl/fl}$ mice (left) and *Lyz2*$^{Cre/+}$.*Egr2*$^{fl/fl}$ mice (right). **g** *Egr2*$^{fl/fl}$ mice (left and filled bar) developed liver fibrosis in 12 weeks, which was ameliorated in liver of *Lyz2*$^{Cre/+}$.*Egr2*$^{fl/fl}$ mice (right and empty bar). Representative images of 10 mice are shown (left). n = 10 mice/genotype. Means ± SEM are shown. *$p$ < 0.05, *t*-test. **f** and **g** scale bars, 200 μm, original magnification, ×10. **h** The amount of collagen was quantitated by the hydroxyproline assay. n = 7–9 mice/genotype. Means ± SEM are shown. *$p$ < 0.05, *t*-test. **i** Fibrosis-associated gene expression was measured by qPCR. n = 7–9 mice/genotype. Mean values relative to *Egr2*$^{fl/fl}$ mice are shown with SEM. *$p$ < 0.05, *t*-test. Each symbol represents an individual animals.

macrophages yielded a gene expression pattern concordant to that of hLAMs (Fig. 6c, top). On the other hand, the gene expression pattern of CD11c$^{lo}$Mincle$^{lo}$ macrophages aligned well with that of moKCs (Fig. 6c, bottom). The differential expression of several hLAM markers (upregulated in CD11c$^{hi}$Mincle$^{hi}$ macrophages) and moKC markers (upregulated in CD11c$^{lo}$Mincle$^{lo}$ macrophages) was verified by qPCR (Supplementary Fig. 5b). By performing scRNA-seq analysis, we were able to show the reduction of cluster 6 and the expansion of cluster 3 in *Lyz2*$^{Cre/+}$.*Egr2*$^{fl/fl}$ mice with MASH (Fig. 4b, e). Consistent with the scRNA-seq data, CD11c and Mincle expression was decreased in *Lyz2*$^{Cre/+}$.*Egr2*$^{fl/fl}$ Tim4$^-$Ly6C$^{lo}$F4/80$^+$ monocyte-derived macrophages (Fig. 6d). Furthermore, the percentage of CD11c$^{hi}$Mincle$^{hi}$ fraction of monocyte-derived macrophages was decreased, whereas that of CD11c$^{lo}$Mincle$^{lo}$ fraction of monocyte-derived macrophages was inversely expanded in *Lyz2*$^{Cre/+}$.*Egr2*$^{fl/fl}$ mice (Fig. 6e).

Microscopically, the majority of CD11c$^{hi}$Mincle$^{hi}$ macrophages corresponding to cluster 6 had abundant pale cytoplasm that was filled with clear vacuoles ("Foamy", Fig. 6f, top and right). On the other hand, CD11c$^{lo}$Mincle$^{lo}$ macrophages corresponding to cluster 3 included a large number of cells that are round in shape with dense nuclei ("Not Foamy", Fig. 6f, bottom and right). CD11c$^{hi}$Mincle$^{hi}$ macrophages contained more neutral lipids than their CD11c$^{lo}$Mincle$^{lo}$ counterparts, as measured by BODIPY staining (Fig. 6g). An enhanced *Cd36*, *Lpl*, and *Fabp5* expression may facilitate lipid uptake[27] by CD11c$^{hi}$Mincle$^{hi}$ macrophages (Fig. 6h). Sorted CD11c$^{hi}$Mincle$^{hi}$ macrophages and CD11c$^{lo}$Mincle$^{lo}$ macrophages were stimulated in vitro with lipopolysaccharide (LPS). This analysis showed that CD11c$^{lo}$Mincle$^{lo}$ macrophages produced much less TNFα than CD11c$^{hi}$Mincle$^{hi}$ macrophages (Fig. 6i), and this may explain why liver fibrosis was alleviated in *Lyz2*$^{Cre/+}$.*Egr2*$^{fl/fl}$ mice. Overall, these findings indicate that Egr2 is required for the differentiation of monocytes into profibrotic, pro-inflammatory hLAMs in MASLD.

**Saturated fatty acids promote *Egr2* expression by infiltrating monocytes**

As shown in Fig. 1h, *Egr2* expression was significantly upregulated in monocytes only after the infiltration into liver, suggesting that some environmental factors accumulated in MASLD liver induce *Egr2* expression by the infiltrating monocytes. Several in vitro studies have defined Egr2 as a signature of IL-4 and/or IL-13-dependent alternative macrophage activation[28–30]. However, our RNA-seq analysis indicated that IL-4 and IL-13 expression was minimal in MASH liver (Supplementary Data 2). We thus set out to determine the environmental factors that trigger *Egr2* expression in the infiltrating monocytes. The dysregulation of lipid metabolism results in the accumulation of several long-chain fatty acids in MASH in both mouse and human[31,32]. Therefore, we evaluated the effect of long-chain saturated fatty acids on Egr2 induction. Saturated fatty acids upregulated *Egr2* mRNA expression in bone marrow-derived macrophages (BMDMs) in a dose-dependent fashion in 6 h (Fig. 7a, top). On the other hand, long-chain unsaturated fatty acids, oleic acid and palmitoleic acid, did not induce *Egr2* expression (Fig. 7a, bottom). Furthermore, addition of unsaturated fatty acids repressed the saturated fatty acid-induced *Egr2*

upregulation (Fig. 7b). Having found that unsaturated fatty acids antagonize the upregulation of *Egr2* by saturated fatty acids, we tested whether unsaturated fatty acids control monocyte differentiation in vivo by administrating oleic acid for 2 weeks to MASH mice. This analysis showed that the percentage of CD11c$^{hi}$Mincle$^{hi}$ hLAMs among Tim4$^-$Ly6C$^{hi}$F4/80$^{lo}$ monocyte-derived macrophages was significantly reduced by oral oleic acid (Fig. 7c). Collectively, these findings suggest that the long-chain saturated fatty acids are one of MASH liver-specific environmental cues that induce Egr2 expression by liver-infiltrating monocytes and that the differentiation into hLAMs can be impeded by oleic acid treatment in MASH.

## Discussion

Recent studies have disclosed that monocyte-derived macrophages are more heterogeneous than previously considered, that is, the recruited macrophages include at least two subsets with distinct activation states that resemble KCs thereby called moKCs and hLAMs[8,11,33]. In this study, we showed that Egr2 played a central role in driving the differentiation of monocytes into hLAMs in MASH liver. In WT mice, infiltrating monocytes gave rise mainly to hLAMs. In *Egr2*-deficient mice, on the other hand, we found that the differentiation of liver-infiltrating monocytes was biased toward moKCs. RNA velocity analysis placed moKCs at the downstream of hLAMs in MASH (Fig. 5a). As *Egr2* expression level was lower in moKCs than in hLAMs (Supplementary Fig. 4e), the downregulation of *Egr2* in hLAMs may be associated with the conversion of hLAMs into moKCs and the subsequent amelioration of liver fibrosis. Our scRNA-seq analysis identified transcriptionally distinct Ly6C$^{hi}$ monocyte subsets, clusters 1 and 4 (Fig. 4b) in MASH liver. Although the RNA velocity analysis did not predict a differentiation of cluster 4 into macrophages, we cannot exclude the possibility that hLAMs and moKCs arise from different monocyte subsets. We noticed that the reduction of hLAMs was not overtly prominent in *Lyz2*$^{Cre/+}$.*Egr2*$^{fl/fl}$ mice compared with *Egr2*$^{fl/fl}$ mice. This may be ascribed to the functional redundancy of another Egr family transcription factor Egr1[34] that was upregulated in *Lyz2*$^{Cre/+}$.*Egr2*$^{fl/fl}$ macrophages when compared to *Egr2*$^{fl/fl}$ macrophages (Supplementary Fig. 3b). Previously, KCs were thought to play an important role in the initiation of MASLD[35,36]. Recently however, roles played by monocyte-derived macrophages are found to be more predominant in the development of liver fibrosis in MASH[13,37]. Overall, we provided evidence that the reduction of hLAMs, a subset of monocyte-derived macrophages, alleviates liver fibrosis in MASH, demonstrating their profibrotic role in vivo. Changes between immune cell types other than monocytes and macrophages, which was not analyzed in detail in this study, may also contribute to the development of liver fibrosis.

By using CD11c and Mincle, we separated the monocyte-derived macrophages into CD11c$^{hi}$Mincle$^{hi}$ hLAMs and CD11c$^{lo}$Mincle$^{lo}$ moKCs. Although we were unable to identify the effector molecule responsible for the promotion of liver fibrosis by CD11c$^{hi}$Mincle$^{hi}$ hLAMs, we found that CD11c$^{hi}$Mincle$^{hi}$ hLAMs produced larger amount of proinflammatory cytokines than CD11c$^{lo}$Mincle$^{lo}$ moKCs. CD11c and Mincle are markers for macrophages constituting a unique structure named crown-like structure

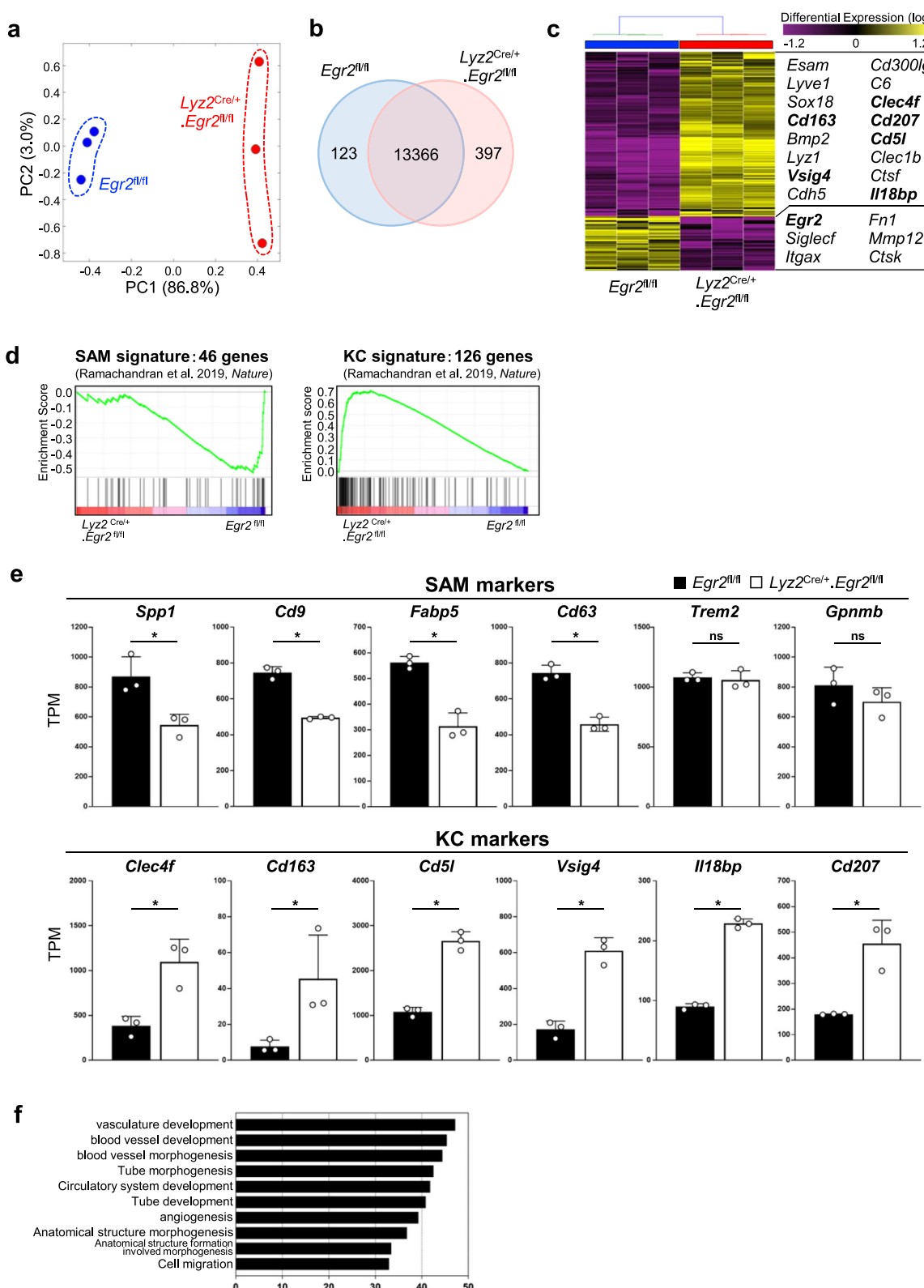

**Fig. 3 | Egr2-deficient macrophages fail to equip SAM characteristics. a** PCA of $Egr2^{fl/fl}$ (blue) and $Lyz2^{Cre/+}.Egr2^{fl/fl}$ (red) macrophages in MASH. Each symbol represents an individual animal. **b** Venn diagram showing the number of genes upregulated (397), unchanged (13,366), or downregulated (123) in $Lyz2^{Cre/+}.Egr2^{fl/fl}$ (pink) macrophages compared with $Egr2^{fl/fl}$ (blue) macrophages. **c** Hierarchical clustering analysis showing the relative expression of discriminative genes in $Egr2^{fl/fl}$ and $Lyz2^{Cre/+}.Egr2^{fl/fl}$ macrophages in MASH liver. Representative DEGs are listed to

the right. **d** GSEA showing the enrichment of markers associated with SAMs (top) or KCs (bottom) in $Egr2^{fl/fl}$ and $Lyz2^{Cre/+}.Egr2^{fl/fl}$ macrophages in MASH liver. **e** Expression levels of known markers of SAMs (top) and KCs (bottom). TPM, transcripts per kilobase million. Filled bars, $Egr2^{fl/fl}$; empty bars, $Lyz2^{Cre/+}.Egr2^{fl/fl}$. Means ± SD are shown. *$p < 0.05$; ns not significant, $t$-test. Each symbol represents an individual animals. **f** GO enrichment of genes upregulated in $Egr2$-deficient macrophages compared with that in $Egr2^{fl/fl}$ macrophages.

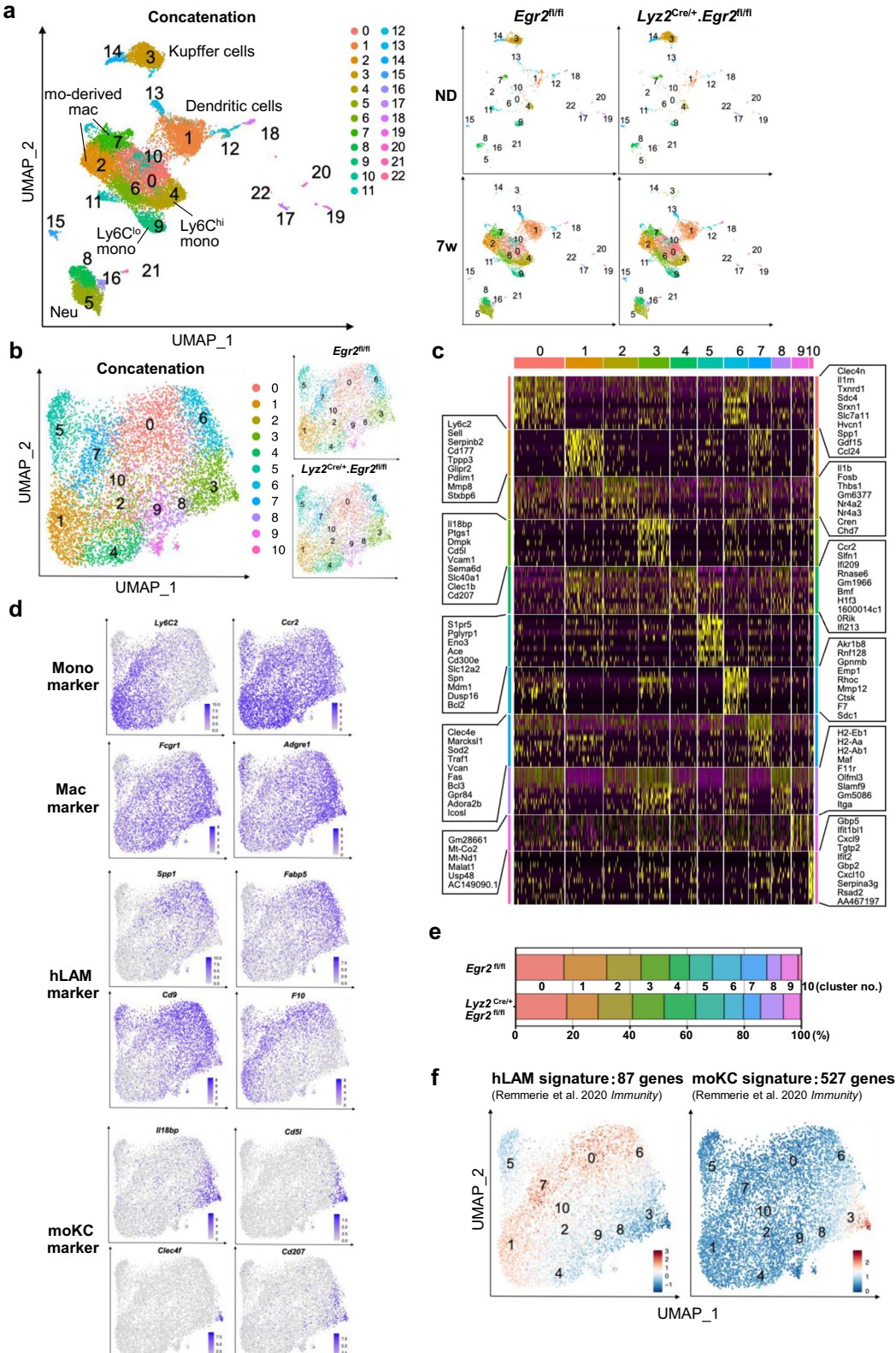

**Fig. 4 | scRNA-seq analysis of liver innate immune cells in WT and *Egr2*-deficient mice. a** UMAP plots of liver CD45.2⁺lin⁻CD11b⁺ cells of ND-fed or CDA-HFD-fed *Egr2*fl/fl mouse and *Lyz2*Cre/+.*Egr2*fl/fl mouse (right). Lin includes CD90.2, B220, NK1.1, and SiglecF. Four UMAP plots are concatenated (left). **b** UMAP plots of reclustered monocytes and mo-derived mac from CDA-HFD-fed *Egr2*fl/fl mouse and *Lyz2*Cre/+.*Egr2*fl/fl mouse (right). Two UMAP plots are concatenated (left).

**c** Heatmap showing top 10 discriminative genes per cluster defined in Fig. 4b. **d** Heatmap showing the expression of monocyte marker genes (*Ly6C2* and *Ccr2*), macrophage genes (*Fcgr1* and *Adgre1*), hLAM marker genes (*Spp1*, *Fabp5*, *Cd9*, and *F10*), and moKC marker genes (*Il18bp*, *Cd5l*, *Clec4f*, and *Cd207*). **e** Proportion of clusters defined by UMAP plots in Fig. 4b. **f** Heatmap showing the calculated module scores for moKC signatures (87 genes) and hLAM signatures (527 genes).

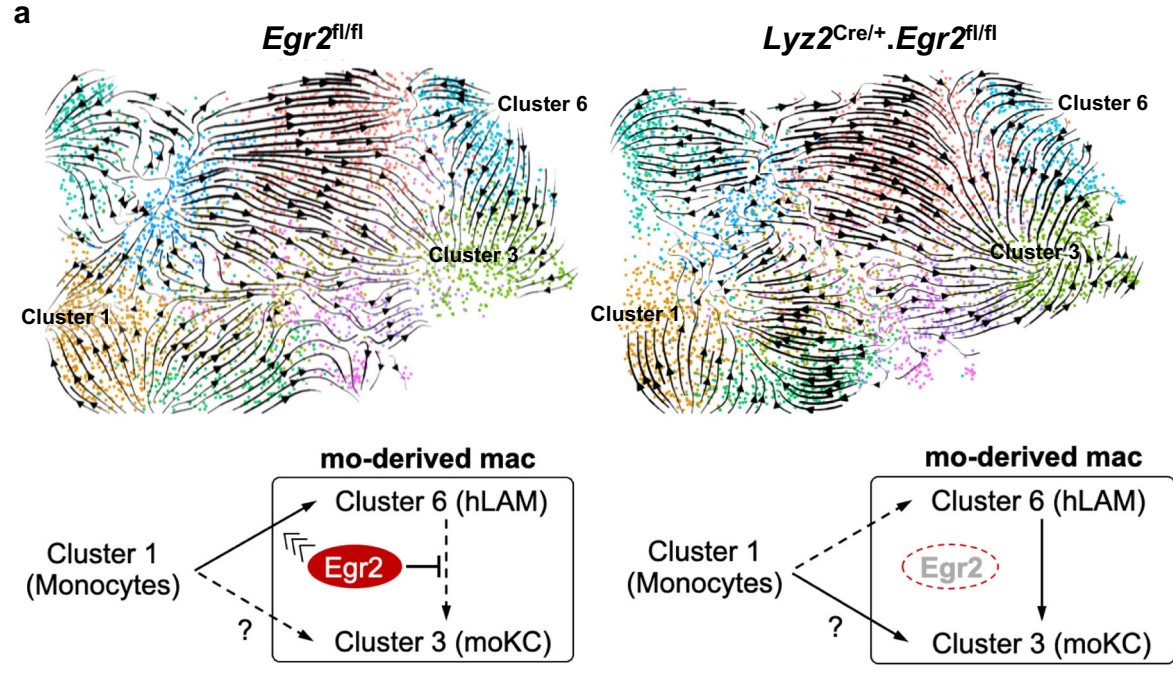

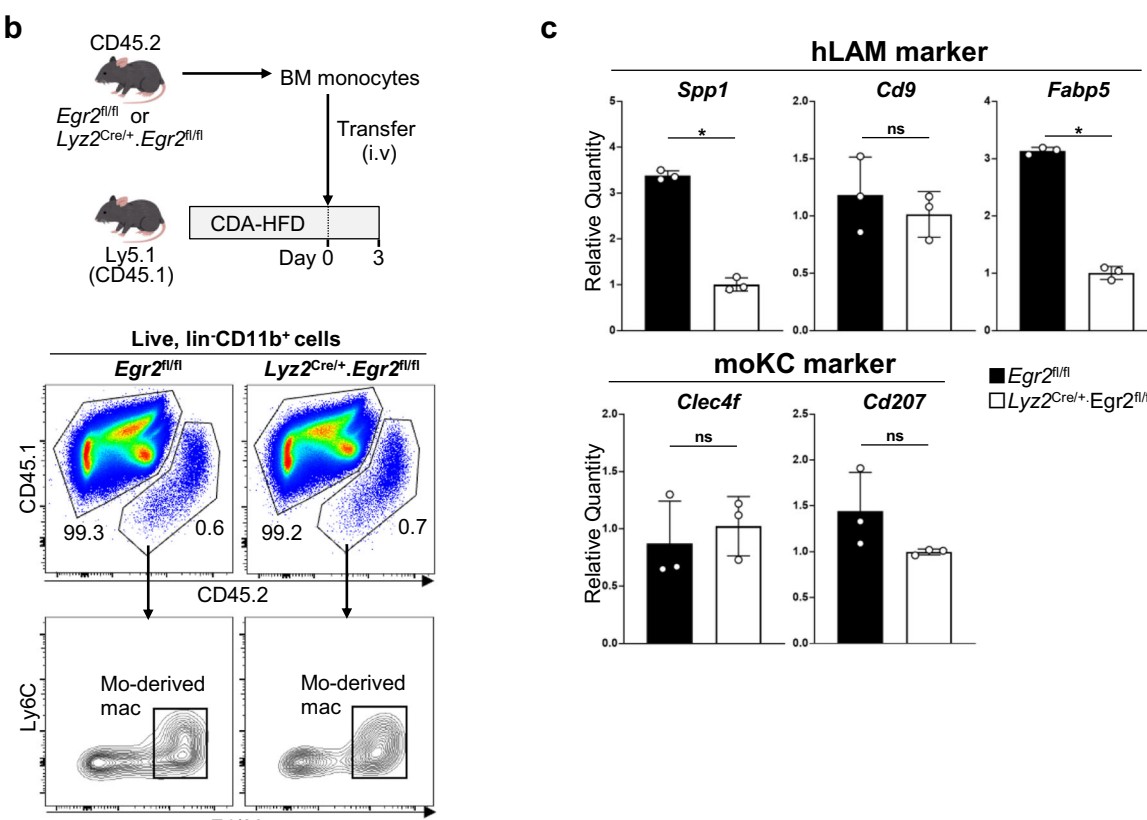

**Fig. 5 | Egr2 favors the differentiation of liver-infiltrating monocytes into hLAMs. a** RNA velocity field projected onto UMAP plots of reclustered monocytes and mo-derived mac from CDA-HFD-fed *Egr2*fl/fl mouse (top left) and *Lyz2*Cre/+.*Egr2*fl/fl mouse (top right) shown in Fig. 4b. Egr2 promotes the differentiation of Ly6Chi monocytes (cluster 1) into hLAMs (cluster 6), and may also inhibit the differentiation of hLAMs into moKCs (cluster 3) in *Egr2*fl/fl mouse (bottom left). In the absence of Egr2, the differentiation of Ly6Chi monocytes (cluster 1) is shifted toward moKCs (cluster 3, bottom right). The conversion of hLAMs into moKCs is also accelerated. **b** Workflow of experimental strategy (top). FACS plots of

CD11b+ cells from recipient liver (CD45.1+) show that *Egr2*fl/fl or *Lyz2*Cre/+.*Egr2*fl/fl CD45.2+ BM monocytes equally differentiated into F4/80+ macrophages in 3 days (bottom). Representative FACS plots of 3 mice/genotype are shown. **c** qPCR analysis of representative hLAM signatures (*Spp1*, *Cd9*, and *Fabp5*) and moKC signatures (*Clec4f* and *Cd207*) in donor mo-derived mac sorted from mice fed CDA-HFD for 7–8 weeks. Average quantities relative to *Lyz2*Cre/+.*Egr2*fl/fl mice are shown. PCR triplicate. Representative results of two experiments are shown. Means ± SD are shown.*$p < 0.05$, *t*-test.

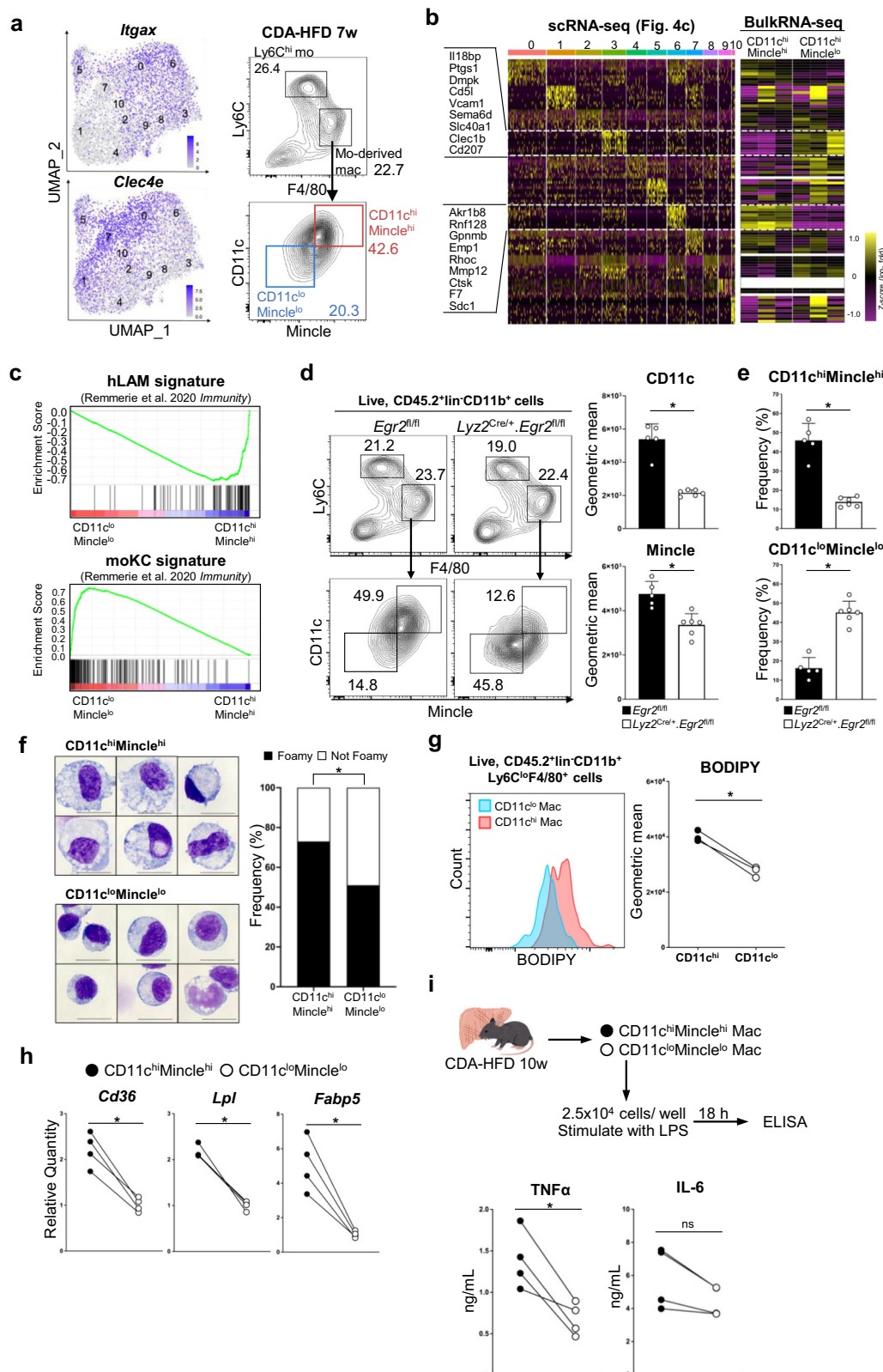

(CLS) in adipose tissue and liver. The CLS-forming macrophages crosstalk with dead hepatocytes, promoting inflammation and fibrosis from MASH[19]. In good agreement with the profibrotic role of CLS-forming macrophages in MASH, we showed that the proportion of CD11c[hi]Mincle[hi] hLAMs was reduced in *Egr2*-deficient mice, which may explain why fibrosis was ameliorated in the absence of *Egr2*. In this study, we could not determine exactly

how Egr2 control the differentiation of hLAMs. As several transcription factors known to regulate monocyte-derived macrophage differentiation were not largely affected by the absence of *Egr2* (Supplementary Fig. 3c), we speculate that Egr2 either directly or indirectly regulate the transcription of hLAM-associated genes without impeding the differentiation of monocytes into macrophages.

**Fig. 6 | CD11c$^{hi}$Mincle$^{hi}$ hLAMs are pro-inflammatory. a** Heatmap showing the expression of *Itgax* (encoding CD11c) and *Clec4e* (encoding Mincle) (left). The proportions of CD11c$^{hi}$Mincle$^{hi}$ macrophages (red) and CD11c$^{lo}$Mincle$^{lo}$ macrophages (blue) in CD45.2$^{+}$lin$^{-}$CD11b$^{+}$Ly6C$^{lo}$F4/80$^{+}$ cells of mice fed CDA-HFD for 7 weeks. Lin includes CD90.2, B220, NK1.1, SiglecF, Ly6G, and Tim4. Representative FACS plots of five mice are shown. **b** Heatmap comparing the relative expression of marker genes discriminating clusters defined in Fig. 4b (left) and in bulk RNA-seq analysis (right) of sorted CD11c$^{hi}$Mincle$^{hi}$ macrophages and CD11c$^{lo}$Mincle$^{lo}$ macrophages from mice fed CDA-HFD for 7 weeks. **c** GSEA showing the enrichment of markers associated with hLAM (top) or moKCs (bottom) in CD11c$^{hi}$Mincle$^{hi}$ macrophages and CD11c$^{lo}$Mincle$^{lo}$ macrophages in MASH liver. **d** FACS plots showing the percentages of liver monocytes and macrophages in CD45.2$^{+}$lin$^{-}$CD11b$^{+}$ cells (left). Lin includes CD90.2, B220, NK1.1, SiglecF, Ly6G, and Tim4. The surface expression of CD11c and Mincle on Ly6C$^{lo}$F4/80$^{+}$ macrophages is decreased in *Lyz2*$^{Cre/+}$.*Egr2*$^{fl/fl}$ mice compared with *Egr2*$^{fl/fl}$ mice (right). Averages of 5-6 mice/genotype are shown with SD, *$p < 0.05$, *t*-test. **e** Bar graphs showing the frequency (%) of CD11c$^{hi}$Mincle$^{hi}$ macrophages (top) and CD11c$^{lo}$Mincle$^{lo}$ macrophages (bottom) among Ly6C$^{lo}$F4/80$^{+}$ macrophages. Averages of 5-6 mice/genotype are shown with SD. *$p < 0.05$, *t*-test. **f** Giemsa

staining showing morphological features of CD11c$^{hi}$Mincle$^{hi}$ macrophages and CD11c$^{lo}$Mincle$^{lo}$ macrophages. CD11c$^{hi}$Mincle$^{hi}$ macrophages contained more lipid droplets than CD11c$^{lo}$Mincle$^{lo}$ macrophages. The frequencies of lipid-rich (foamy, black) and lipid-poor (not foamy, white) macrophages among more than 500 cells counted. *$p < 0.05$, Pearson's chi-square test. Scale bars represent 20 μm. **g** Neutral lipid content of mo-derived mac from mice-fed CDA-HFD for 7 weeks was measured by flow cytometry (left). CD11c$^{hi}$ macrophages (red) contained more neutral lipids than CD11c$^{lo}$ macrophages (blue). Geometric mean of BODIPY fluorescence intensity (right). Each symbol represents an individual animal. *$p < 0.05$, paired *t*-test. **h** Gene expression of indicated lipid uptake markers (*Cd36*, *Lpl*, and *Fabp5*) was measured by qPCR. n = 4 mice/cell type. Average quantities relative to CD11c$^{lo}$Mincle$^{lo}$ macrophages are shown. Each symbol represents an individual animal. *$p < 0.05$, paired *t*-test. (**i**) Concentrations of pro-inflammatory cytokines (TNFα and IL-6) in the culture medium of CD11c$^{hi}$Mincle$^{hi}$ macrophages and CD11c$^{lo}$Mincle$^{lo}$ macrophages that were stimulated with LPS (0.1 μg/mL) overnight were quantified by ELISA. Each symbol represents an individual animal. n = 4/ cell type. *$p < 0.05$; ns not significant; paired *t*-test. **d**, **e**, **h** and **i** Each symbol represents an individual animal.

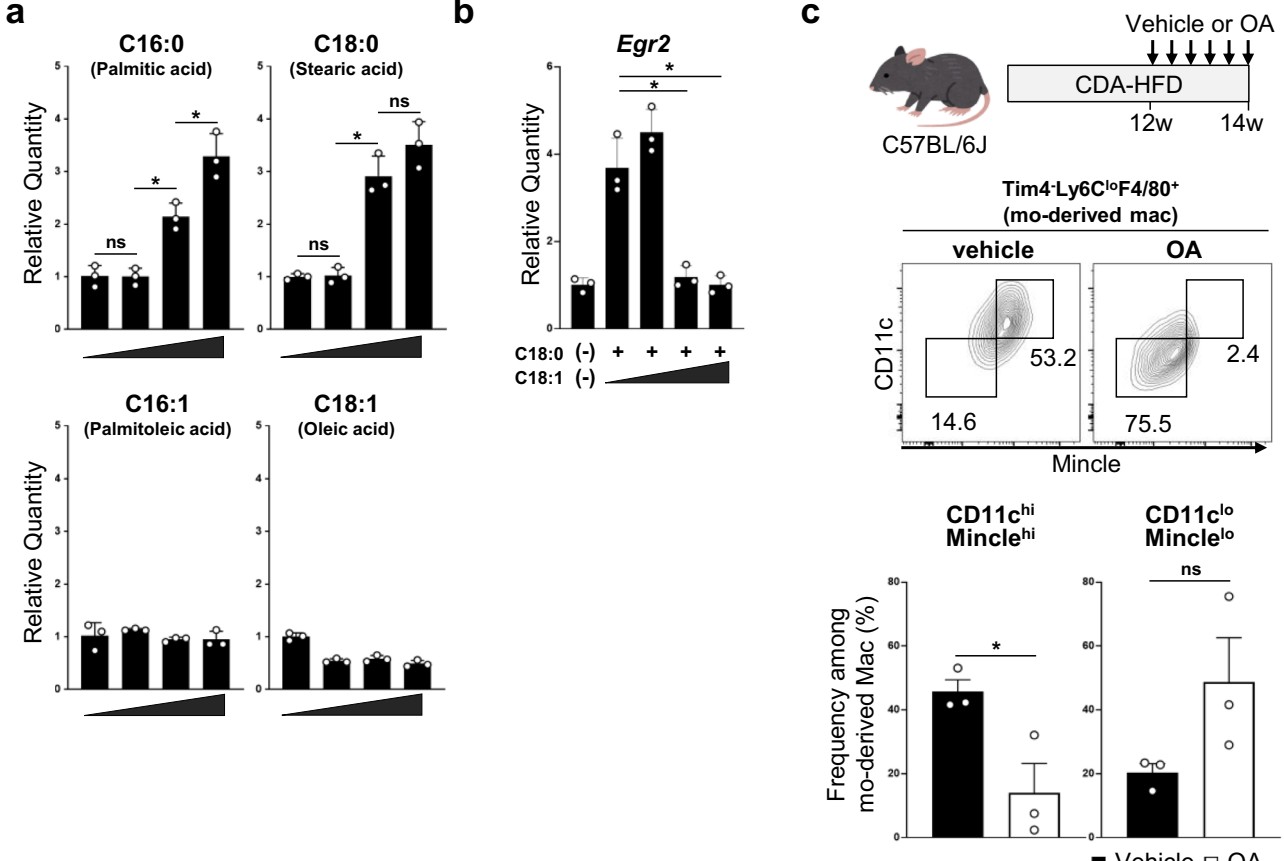

**Fig. 7 | Saturated fatty acids stimulate the expression of *Egr2* in macrophages. a** qPCR analysis of *Egr2* mRNA levels of BMDMs that were stimulated with saturated fatty acids (C16:0, palmitic acid and C18:0, stearic acid, top) or unsaturated fatty acids (C16:1, palmitoleic acid and C18:1, oleic acid, bottom) at the concentration of 0, 25, 100, or 400 μM for 6 h. **b** Oleic acid repressed the expression of *Egr2* induced by saturated acid. BMDMs were stimulated with 200 μM saturated fatty acid (C18:0) in the presence of 0, 100, 200, or 400 μM unsaturated fatty acid (C18:1) for 6 h. (-)

indicates vehicle control. **a** and **b** n = 3/condition. Experiments are performed twice in triplicate. Means ± SD are shown. *$p < 0.05$; ns not significant, one-way ANOVA. **c** MASH mice were orally administered with vehicle or oleic acid (OA) for 2 weeks. Percentage of CD11c$^{hi}$Mincle$^{hi}$ hLAMs among monocyte-derived macrophages was reduced by oleic acid (OA) administration in MASH mice. n = 3 mice. Means ± SEM are shown. *$p < 0.05$, *t*-test. Each symbol represents an individual animal.

Once monocytes arrive at their tissue of destination, they undergo extensive differentiation depending on the environmental cues provided by tissue-specific niches, enabling their development into specialized macrophages that support local tissue function[38,39]. Following non-inflammatory KC depletion, Notch ligand expressed by liver sinusoidal endothelial cells

(LSECs) rapidly activates poised enhancers to induce KC lineage-determining factor LXRα in monocytes[40]. In intact liver, hepatocyte-derived desmosterol, a cholesterol precursor, and LSEC-derived TGF-β coordinately convert infiltrating monocytes into Kupffer-like cells in 2 weeks. However, environmental factors that initiate the differentiation

## Table 1 | Antibodies

| Antibodies | Source | Clone |
|---|---|---|
| Brilliant Violet 510™ anti-mouse CD45.1 | BioLegend | A20 |
| PE anti-mouse CD45.2 | BioLegend | 104 |
| Brilliant Violet 421™ anti-mouse CD45.2 | BioLegend | 104 |
| FITC anti-mouse CD45.2 | BioLegend | 104 |
| Biotin anti-mouse CD45.2 | BioLegend | 104 |
| Brilliant Violet 421™ anti-mouse/human CD11b | BioLegend | M1/70 |
| APC/Cyanine7 anti-mouse/human CD11b | BioLegend | M1/70 |
| Alexa Fluor® 488 anti-mouse Ly-6C | BioLegend | HK1.4 |
| PE anti-mouse Ly-6C | BioLegend | HK1.4 |
| PE/Cyanine7 anti-mouse F4/80 | BioLegend | BM8 |
| APC/Cyanine7 anti-mouse Ly-6G | BioLegend | 1A8 |
| PerCP/Cyanine5.5 anti-mouse Ly-6G | BioLegend | 1A8 |
| PerCP/Cyanine5.5 anti-mouse CD90.2 (Thy1.2) | BioLegend | 53-2.1 |
| PerCP anti-mouse CD90.2 (Thy1.2) | BioLegend | 30-H12 |
| PerCP/Cyanine5.5 anti-mouse/human CD45R/B220 | BioLegend | RA3-6B2 |
| PerCP-Cy™5.5 Rat Anti-Mouse CD45R/B220 | eBioScience | RA3-6B2 |
| PerCP/Cyanine5.5 anti-mouse NK-1.1 | BioLegend | PK136 |
| PerCP-Cy™5.5 Rat Anti-Mouse Siglec-F | eBioScience | E50-2440 |
| PerCP/Cyanine5.5 anti-mouse CD170 (Siglec-F) | BioLegend | S17007L |
| APC anti-mouse Tim-4 | BioLegend | RMT4-54 |
| PerCP/Cyanine5.5 anti-mouse Tim-4 | BioLegend | RMT4-54 |
| Alexa Fluor® 647 anti-mouse Tim-4 | BioLegend | F31-5G3 |
| APC anti-mouse CD11c | BioLegend | N418 |
| PE anti-mouse CD19 | BioLegend | 6D5 |
| APC anti-mouse CD3ε | BioLegend | 145-2C11 |
| APC anti-mouse CD115 (CSF-1R) | BioLegend | AFS98 |
| Anti-Mincle (Mouse) mAb-Biotin | MBL | 1B6 |
| Purified anti-mouse Tim4 | BioLegend | RMT4-54 |
| Biotin anti-mouse F4/80 | BioLegend | CI;A3-1 |

program of monocyte-derived macrophages in MASLD were not identified. Comprehensive lipidomic analysis of MASH patient livers specified a signature of lipid species discriminating MASH patients from healthy or steatosis patients[32]. That study revealed the accumulation of long-chain fatty acids as a result of dysregulated FADS1 desaturase activity with 100% specificity and sensitivity in MASH patients. Here, we showed that long-chain saturated fatty acids but not their unsaturated counterparts induced *Egr2* expression in monocytes and macrophages. Even more, we found that one of unsaturated fatty acids, oleic acid downregulated *Egr2* level in saturated-fatty acid-treated macrophages. How monocytes discriminate saturated or unsaturated fatty acids in the context of *Egr2* transcription should be determined. Strikingly, oral administration of oleic acid reduced the proportion of hLAMs in monocyte-derived macrophages in only 2 weeks. Although we demonstrated that the oral administration of oleic acid inhibits the differentiation of monocytes into hLAMs in MASH mice, long-term suppression of liver fibrosis was not examined in this study. It is intriguing whether oral oleic acid therapy can suppress the progression of liver fibrosis from simple steatosis in MASH.

*Egr2* is upregulated not only in CDA-HFD-induced MASH model but also in HFD- and AMLN-induced MASH models[9,11]. Ramachandran's group reported that *Egr2* level is enhanced in liver macrophages of human MASH patients[13]. These reports suggest that Egr2 plays a key role in promoting the differentiation of monocytes into hLAMs both in mouse and human.

Egr2 is indispensable for maintaining the unique functions and characteristics of alveolar macrophages[41]. In lung, Egr2 is induced by local GM-CSF and TGF-β in a PPAR-γ-dependent manner in infiltrating monocytes. These findings suggest that the upregulation of Egr2 driven by a niche-specific factor is a conserved monocyte-derived macrophage response that is present across different tissues. In T cells, Egr2 plays an important role for the maintenance of T cell anergy[42]. LAG3+CD4+CD25- regulatory T cells produce a large amount of TGF-β3 in an Egr2-dependent manner, which is responsible for the suppression of systemic humoral immune response in a mouse model of lupus[43]. Thus, although Egr2 might be a target for liver fibrosis in MASH, Egr2 should be targeted selectively in monocytes and macrophages in order not to inhibit protective responses by LAG3+CD4+CD25- regulatory T cells.

## Methods

### Mice

Six- to twelve-week-old C57BL/6J mice and CD45.1 congenic mice were purchased from CLEA Japan and Sankyo Labo Service, respectively. LyzM-Cre mice (*Lyz2*-Cre) were provided by RIKEN BRC[44]. *Egr2*-flox mice were kindly provided by Dr. Charnay[20]. All mice were bred in specific pathogen-free animal facilities. We have compiled with all relevant ethical regulations for animal use.

### Mouse model of MASH

Male mice (6- to 12-week-old) were fed choline-deficient, L-amino acid-defined high-fat diet (CDA-HFD, Research Diets, NJ, U.S.A.) containing 0.1% methionine and 62% kcal fat. In some experiments, mice were given 200 mg/kg oleic acid in 250 μL distilled water or vehicle control by oral gavage 3 times a week for 2 weeks. Serum AST and ALT concentrations were determined enzymatically with a biochemical analyzer, DRI-CHEM NX500sV (Fujifilm).

### Purification of spleen lymphocytes

Spleen was squeezed between glass slides, and the cell suspension was filtered through nylon mesh. Red blood cells were lysed with 1x Pharm Lyse (BD Biosciences) for 2 min. Cells were pre-incubated with Fc-blocker (Clone 93, BioLegend) for 5 min on ice and then stained with a mixture of antibodies against surface proteins (detailed in Table 1) for 30 min on ice. Dead cells were excluded by using 7-AAD (Miltenyi Biotech). T cells and B cells were sorted with a FACSAria III (BD Biosciences) cell sorter.

### Isolation of liver leukocytes

Mice were deeply anesthetized with a mixture of medetomidine (0.3 mg/kg weight), midazolam (4 mg/kg weight), and butorphanol (5 mg/kg weight). The liver was perfused with 25 mL of pre-warmed Liver Perfusion Medium (Thermo Fisher Scientific) followed by 25 mL of pre-warmed collagenase perfusion solution (0.5 mg/mL Collagenase IV (Gibco), 136 mM NaCl, 5.4 mM KCl, 5 mM CaCl$_2$, 0.5 mM NaH$_2$PO$_3$, 0.42 mM Na$_2$HPO$_3$, 10 mM HEPES, pH 7.5, 5 mM glucose, and 4.2 M NaHCO$_3$) at a rate of 3 mL/min with a Perista pump (ATTO, Japan) through the portal vein. The liver was carefully removed, minced, and further digested in 25 mL of collagenase perfusion solution for 25 min at 37 °C under gentle rotation. After the enzymatic digestion, cells were passed through a 70-μm cell strainer (BD Biosciences). Hepatocytes were removed by three to five cycles of centrifugation at $100 \times g$ for 2 min. Non-parenchymal cells including leukocytes in the supernatant were centrifuged in 50 mL of serum-free DMEM at $500 \times g$ for 10 min at 4 °C before proceeding to antibody staining for flow cytometry.

### In vivo transfer of BM monocytes

BM monocytes were enriched by magnetic sorting with Monocyte Isolation Kit (Miltenyi Biotech) and autoMACSpro separator (Miltenyi Biotech) from naïve CD45.2+ *Lyz2*Cre/+.*Egr2*fl/fl mice or *Egr2*fl/fl mice. Those monocytes were transferred intravenously into CD45.1+ MASH-induced mice.

## Table 2 | PCR primer sequences

| Gene name | | Sequence(5'→3') |
|---|---|---|
| qPCR primers | | |
| Rn18s | Fwd | CGGACAGGATTGACAGATTG |
| | Rev | CAAATCGCTCCACCAACTAA |
| Egr2 | Fwd | CTACCCGGTGGAAGACCTC |
| | Rev | AATGTTGATCATGCCATCTCC |
| Timp1 | Fwd | GCAAAGAGCTTTCTCAAAGACC |
| | Rev | AGGGATAGATAAACAGGGAAACACT |
| Col1a1 | Fwd | CATGTTCAGCTTTGTGGACCT |
| | Rev | GCAGCTGACTTCAGGGATGT |
| Acta2 | Fwd | GACACCACCCACCCAGAGT |
| | Rev | ACATAGCTGGAGCAGCGTCT |
| Tgfb1 | Fwd | GGACTCTCCACCTGCAAGAC |
| | Rev | GACTGGCGAGCCTTAGTTTG |
| Tnf | Fwd | TCTTCTCATTCCTGCTTGTGG |
| | Rev | GGTCTGGGCCATAGAACTGA |
| Il1b | Fwd | GGATGAGGACATGAGCACCT |
| | Rev | AGCTCATATGGGTCCGACAG |
| Spp1 | Fwd | GGAAACCAGCCAAGGTAAGC |
| | Rev | TGCCAATCTCATGGTCGTAG |
| Cd9 | Fwd | GATATTCGCCATTGAGATAGCC |
| | Rev | TGGTAGGTGTCCTTGTAAAACTCC |
| Fabp5 | Fwd | TGAAAGAGCTAGGAGTAGGACTG |
| | Rev | CTCTCGGTTTTGACCGTGATG |
| Trem2 | Fwd | TGGGACCTCTCCACCAGTT |
| | Rev | GTGGTGTTGAGGGCTTGG |
| Mmp12 | Fwd | TCAATTGGAATATGACCCCCTG |
| | Rev | ACCAGCAAGCACCCTTCACTAC |
| F7 | Fwd | AATGAGCAGCTGATCTGTGC |
| | Rev | GCAGGACACCTCATCTGGCT |
| Atp6v0d2 | Fwd | AAGCCTTTGTTTGACGCTGT |
| | Rev | TGAATGCCAGCACATTCATC |
| Ctsk | Fwd | GCAGAGGTGTGTACTATGA |
| | Rev | GCAGGCGTTGTTCTTATT |
| Cd36 | Fwd | GGAGCCATCTTTGAGCCTTCA |
| | Rev | GAACCAAACTGAGGAATGGATCT |
| Lpl | Fwd | ACAAGGTCAGAGCCAAGAGAAG |
| | Rev | TGGTTGTGTTGCTTGCCATC |
| Cd207 | Fwd | CCGAAGCGCACTTCACAGT |
| | Rev | GCAGATACAGAGAGGTTTCCTTA |
| Clec1b | Fwd | GTGATGGCTTTAGTTCTGCTGAT |
| | Rev | CTTTTGCTGTGTGACCGACA |
| Clec4f | Fwd | GAGGCCGAGCTGAACAGAG |
| | Rev | TGTGAAGCCACCACAAAAGAG |
| Cd163 | Fwd | CCTGGATCATCTGTGACAACA |
| | Rev | TCCACACGTCCAGAACAGTC |
| Nr1h3 | Fwd | GAAATGCCAGGAGTGTCGAC |
| | Rev | AAGCGGATCTGTTCTTCTGACAG |
| Atf3 | Fwd | CTCTGCCATCGGATGTCCTC |
| | Rev | GTTTCGACACTTGGCAGCAG |
| Bhlhe40 | Fwd | GACCGGATTAACGAGTGCAT |
| | Rev | TGCTTTCACGTGCTTCAACG |

## Table 2 (continued) | PCR primer sequences

| Gene name | | Sequence(5'→3') |
|---|---|---|
| qPCR primers | | |
| Pparg | Fwd | TGTGGGGATAAAGCATCAGGC |
| | Rev | CCGGCAGTTAAGATCACACCTAT |
| Rbpj | Fwd | ATGGACTACTCGGAGGGCTT |
| | Rev | AGCACTGTTTGATCCCCTCG |
| Nfil3 | Fwd | GAACTCTGCCTTAGCTGAGGT |
| | Rev | ATTCCCGTTTTCTCCGACACG |
| Genotyping primers | | |
| Lyzs_pro | | GCATTGCAGACTAGCTAAAGGCAG |
| Lyzs_ex1_r | | GTCGGCCAGGCTGACTCCATAG |
| Cre8 | | CCCAGAAATGCCAGATTACG |
| Egr2_flox_p4 | | GGGAGCGAAGCTACTCGGATACGG |
| Egr2_flox_p5 | | GTTGCTCTGTAGTGTTGGAATCATG |

Differentiation of transferred monocytes in the recipient liver was analyzed 3 days after the injection.

### Flow cytometry of liver leukocytes

For flow cytometric measurements, cells were pre-incubated with Fc-blocker (Clone 93, BioLegend) for 5 min on ice and then stained with a mixture of antibodies against surface proteins (detailed in Table 1) for 30 min on ice. Dead cells were excluded by using DAPI (Dojindo) or 7-AAD (Miltenyi Biotech). In some experiments, CD45.2$^+$ leukocytes were enriched by magnetic sorting using autoMACSpro (Miltenyi Biotech) before flow cytometry. Cell suspensions were analyzed with a BD FACSCelesta or a FACSAria III (BD Biosciences) cell sorter. Data analysis was performed using FlowJo v10 software (Becton Dickinson and Company).

### Quantitative real-time PCR (qPCR)

Total RNA from the sorted cells was extracted with an RNeasy Mini Kit, Micro Kit (QIAGEN, Netherlands), or a FavorPrep Total RNA Extraction Column (Favorgen, Taiwan) according to the manufacturers' protocols. For qPCR, complementary DNAs (cDNAs) were synthesized by using ReverTra Ace (TOYOBO, Japan). qPCR was performed on cDNA using a THUNDERBIRD SYBR qPCR Mix (Toyobo, Japan). Expression levels were normalized to 18S ribosomal RNA (rRNA). Primer sequences are summarized in Table 2.

### Quantitation of TNFα and IL-6 concentrations

Sorted liver macrophages were stimulated in vitro with 0.1 µg/mL LPS (O55:B5, Sigma) for 24 h. TNFα and IL-6 concentrations in the culture medium were determined by ELISA MAX™ Standard Set Mouse (BioLegend) according to the manufacturer's protocols.

### Hepatic triglyceride level

Total lipids in the liver were extracted with ice-cold 2:1 (vol/vol) chloroform/methanol. Hepatic triglyceride (TG) concentrations were determined using LabAssay™ Triglyceride (FUJIFILM Wako, Japan).

### Hydroxyproline assay

Hydroxyproline content was measured by using a Hydroxyproline Assay Kit (Sigma) according to the manufacturer's protocol. In brief, ~100 mg of minced liver was homogenized in a homogenizer (Shakeman 6, BMS, Japan). The homogenate was hydrolyzed with 6 N HCl at 110–120°C for 4 h. After the addition of activated charcoal powder and centrifugation at $20,000 \times g$ for 3 min, 12.5 µL of supernatant was transferred onto a 96-well plate that was placed over a 60 °C-heat block to dry the samples. Fifty microliters of chloramine T/oxidation buffer mixture was added to each

well, and incubation was carried out at RT for 5 min. Then, an equal volume of diluted DMAB reagent was added, and incubation was accomplished at 60 °C for 30 min. The absorbance at 560 nm was measured with an iMark™ Microplate Reader (Bio-Rad, U.S.A.).

## Histopathology

Liver tissue was fixed in 10% formaldehyde and embedded in paraffin. 4-μm-thick sections were prepared by a microtome and mounted on silanized glass slides. De-paraffinized sections were stained with either hematoxylin and eosin (H&E) or picrosirius red solution (0.1% Direct Red 80 [Sigma-Aldrich, Missouri, U.S.A.] in saturated aqueous picric acid [Muto chemical, Japan]). The sections were then serially immersed in 70%, 80%, 90%, and 100% ethanol, permeabilized in xylene, and mounted in Marinol (Muto Chemical, Japan). The stained sections were observed under a light microscope (BZ-X700, KEYENCE, Japan). Sirius red positive area was quantitated by using ImageJ Fiji software.

## Immunohistochemistry

For the visualization of F4/80 and Tim4, 8-μm-thick fresh frozen sections were immersed in 0.5% $H_2O_2$/methanol to quench endogenous peroxidase activity. Sections were then blocked with Biotin-Blocking System (DAKO) and then with 0.1 M Tris-HCl, pH 8.0/150 mM NaCl/0.05% Tween x20 (TN blocking buffer). Those sections were incubated with either biotinylated anti-F4/80 (2 μg/mL, BioLegend, Clone CI:A3-1), or rat IgG anti-Tim4 (2 μg/mL, BioLegend, Clone RMT4-54) and biotinylated-anti-rat IgG (×250, proteintech), in TN blocking buffer. Signals were amplified by using TSA Biotin System (AKOYA Biosciences). Biotin-conjugated antigens were detected by Alexa Fluor 750-conjugated streptavidin (Invitrogen). Nuclei were visualized by DAPI. Stained sections were mounted in FluorSave (Millipore) and observed under a fluorescent microscope (BZ-X700, KEYENCE, Japan).

## In vitro induction of Egr2 in BMDMs

Femurs and tibias of C57BL/6 mice were used for BM cell collection. BM cells were cultured in DMEM supplemented with 10% FBS/10% CMG14-12 conditioned medium/100 U/mL penicillin-streptomycin (Thermo Fisher Scientific). The medium was replaced on day 3 to remove non-adherent cells. BMDMs were harvested on day 5 or 6, seeded on a 96-w plate, and cultured for another 24 h. BMDMs were stimulated with 0–400 μg/mL of palmitic acid (Cayman), palmitoleic acid (Tokyo Chemical Industry), stearic acid (Fujifilm), or oleic acid (Fujifilm) for 6 h. Then, BMDMs were lysed with Buffer RLT (Qiagen) or FARB buffer (Favorgen) for total RNA extraction.

## RNA-sequencing analysis

Total RNA from sorted cells was extracted with either TRIzol LS reagent (Thermo Fisher Scientific), Favorgen RNA Kit (Favorgen), or RNeasy Micro Kit (QIAGEN). cDNA was synthesized and amplified using a SMART-Seq HT Kit (Clontech). RNA-seq libraries were prepared using a Nextera XT Kit (Illumina). Single-end 75-bp sequencing was conducted on a NextSeq 500 platform (Illumina). For low-input RNA-seq analysis of $CD11c^{hi}Mincle^{hi}$ $Ly6C^{lo}F4/80^+$ macrophages and $CD11c^{lo}Mincle^{lo}$ $Ly6C^{lo}F4/80^+$ macrophages, cDNA was synthesized and amplified using an NEBNext Poly(A) mRNA Magnetic Isolation Module (New England BioLabs). RNA-seq libraries were prepared using an NEBNext Ultra II Directional RNA Library Prep Kit (New England BioLabs). Paired-end 150-bp sequencing was conducted on an Illumina NovaSeq 6000 platform (Illumina).

Adaptor sequences were removed from raw data using Trimmomatic (ver. 0.36 or 0.39). Sequencing reads after trimming were aligned to a mouse reference genome sequence (GRCm38/mm10) in HISAT2 (ver. 2.2.1). PCR duplicates were removed by using Picard (ver.2.23.9). Read counts per gene were calculated using Cufflinks (ver. 2.1.1) or FeatureCounts (ver. 2.0.1) followed by conversion into transcripts per kilobase million (TPM) based on UCSC mm10. PCA and hierarchical clustering analysis were performed

AltAnalyze software[45]. GO enrichment analysis was performed with the PANTHER Classification System[46]. Gene set enrichment analysis was performed using GSEA software (ver. 4.1.0). Standard parameters with gene set permutation type were used for the analysis and differences were considered significant with FDR < 0.25. Gene sets used in this project are listed in Supplementary Data 3.

## scRNA-seq analysis

$CD45.2^+$Lin (CD90.2, B220, NK1.1, SiglecF)⁻$CD11b^+$ liver cells from ND-fed or CDA-HFD-fed *Egr2*fl/fl mouse or *Lyz2*Cre/+.*Egr2*fl/fl mouse (one mouse per group) were sorted by FACSAriaIII. Each cell sample was stained with biotin anti-mouse CD45.2 and with different DNA barcode-conjugated PE-streptavidins (BioLegend TotalSeq PE-Streptavidin [A951, A952, A953, and A954]) to identify its origin. Then, the sorted cells were pooled and subjected to scRNA-seq using a BD Rhapsody™ Single-Cell Analysis System (BD Biosciences), and resultant cDNA was amplified by TAS-Seq protocol as previously described[47] by ImmunoGeneTeqs Inc. Briefly, on-beads cDNA was poly C tailed under stochastic termination condition by terminal transferase, deoxycytidine, and spiked-in dideoxycytidine. Then, second-strand synthesis and whole-transcriptome amplification were performed by PCR. Size distribution of cDNA and TotalSeq PE-streptavidin libraries was analyzed by a MultiNA system (Shimadzu). The resultant cDNA library was processed into a sequencing library by using an NEBNext Ultra II FS DNA Library Prep Kit for Illumina (New England Biolabs), and sequencing adapters were added to associated TotalSeq PE-streptavidin libraries by PCR. Sequencing was performed by an Illumina NovaSeq 6000 sequencer (Illumina, San Diego, CA, USA) and a NovaSeq 6000 S4 Reagent Kit v1.5 (200 cycles). Pooled library concentration was adjusted to 2.0 nM, and 12% PhiX control library v3 (Illumina) was spiked into the library. The sequencing configuration is as follows: read1 67 base pairs [bp], read2 151 bp, index1 8 bp, and index2 8 bp. Adapter trimming, quality filtering, and mapping to the cell barcode and GRCm38-101 reference transcriptome or TotalSeq PE-streptavidin reference of fastq files were performed by using a previously described pipeline (https://github.com/s-shichino1989/TAS-Seq)[47]. Demultiplexing of scRNA-seq data by TotalSeq PE-streptavidin was performed by using a previously described pipeline (https://github.com/s-shichino1989/TASSeq)[47].

The resultant gene-expression count matrix was processed for downstream single-cell analyses (integration of four datasets [A951, A952, A953, and A954], UMAP dimension reduction, cell cluster identification, conserved marker identification) using Seurat ver. 4.1.1 in R ver. 4.2.1[48]. Briefly, A951, A952, A953, and A954 datasets were integrated with merge functions of Seurat. Cells that contained a percentage of mitochondrial transcripts >25% were filtered out. PCA analysis was performed against 5054 of highly-variable genes identified by FindVariableFeatures (selection.method = mvp, mean.cutoff = c(0.1, Inf), dispersion.cutoff = c(0.5, Inf)) function in Seurat. 1:57 PCs were selected by Jackstraw analysis and used for clustering analysis. Resolution was set as 1.0 for the FindCluster function. UMAP was performed on significant PCs for visualization in two dimensions. Next, marker genes in each cell cluster were defined by FindAllMarkers function in Seurat (test method = wilcox, minimum expression in each cluster R 20%). In sub-clustering analysis of cluster 0,2,4,6,7,9,10, and 11, PCs were set to 1:95. The resolution of FindCluster function was set to 1.0. Average gene expression in each cluster was calculated by AverageExpression function in Seurat. To assess the characteristics of each macrophage cluster, the module score was calculated with Addmodule score function[24] of Seurat package. Gene sets used in this project are listed in Supplementary Data 4.

## RNA velocity analysis

For RNA velocity analysis, cDNA reads were mapped to reference genome (build GRCm38_101) as described previously[49]. Then, BAM file was splitted by valid cell barcodes by using nim ver.1.0.6 and hts-nim ver.0.2.23. Resultant BAM files were processed into loom files by using velocyto run with -c and -U options, and the loom files were concatenated by

loompy.combine function. RNA velocity estimation of monocyte/macrophages in $Egr2^{fl/fl}$ mouse or $Lyz2^{Cre/+}.Egr2^{fl/fl}$ mouse was performed by using scVelo[25] with reticulate package in R ver.4.2.1. Briefly, gene filtering and normalization of $Egr2^{fl/fl}$ and $Lyz2^{Cre/+}.Egr2^{fl/fl}$ data was performed by filter_and_normalize function (min_shared_counts=as.integer (30), min_shared_cells=5), and moments were calculated pp.moments function (n_pcs=35, n_neighbors=30, use_highly_variable=FALSE). Velocity graph was calculated by using tl.velocity function (mode='dynamical', use_highly_variable=FALSE) and tl.veelocity_graph function, and visualized by pl.velocity_embedding_stream function (min_mass = 2, size = 15).

## Evaluation of lipid uptake by liver macrophages

The morphology of sorted liver $CD11c^{hi}Mincle^{hi}$ $Ly6C^{lo}F4/80^+$ macrophages and $CD11c^{lo}Mincle^{lo}$ $Ly6C^{lo}F4/80^+$ macrophages was analyzed by Giemsa staining. In brief, 20,000–25,000 sorted macrophages were cytospun on a glass slide and stained with Diff-Quick (Dade Behring) according to the manufacturer's protocol. The percentage of cells containing lipid inclusions was determined in high-power fields by two independent scientists (A.I. and K.A.) who were blinded to the identities of the sample. Statistical significance was assessed by the chi-square test. BODIPY staining (Thermo Fisher) was performed after surface antibody staining for 15 min at 37 °C, and analysis was performed by FACSCelesta (BD).

## Statistics and reproducibility

Data were analyzed either by analysis of variance (ANOVA) followed by multiple comparison, or by the paired or unpaired $t$-test with Prism (GraphPad Software) unless otherwise stated. $P$ values < 0.05 were considered significant.

## Study approval

All animal experiments were performed in accordance with applicable guidelines and regulations and were approved by the Tokyo University of Pharmacy and Life Sciences (TUPLS) animal use committee (Approval numbers: L19-21, L20-18, L21-18, L21-19, L22-17, L22-18, and L23-08).

## Reporting summary

Further information on research design is available in the Nature Portfolio Reporting Summary linked to this article.

## Data availability

Bulk RNA-seq and scRNA-seq data generated for this study have been deposited at GEO and are publicly accessible using accession numbers GSE263970, GSE263972, GSE263973, GSE263974, and GSE263975. Additional supplementary files include Supplementary Data 1–4. Source data behind the graphs in the paper can be found in Supplementary Data 5.

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

## Acknowledgements

This study was supported in part by Grant-in-Aid for Transformative Research Areas (A) (22H05190 to K.A.), Grants-in-Aid for Scientific Research (B) (20H03473 to Masato, T.), Grants-in-Aid for Scientific Research (C) (21K06877 to K.A.), Grants-in-Aid for Challenging Exploratory Research (21K19387 to Masato, T.), Grant-in-Aid for Transformative Research Areas (B) (22H05064 to SS), the Japan Agency for medical Research and Development (AMED)-CREST (JP18gm1210002 to Masato, T. and Minoru, T.), AMED-PRIME (JP21gm6210025 to S.S.), MEXT Promotion of Distinctive Joint Usage/Research Center Support Program at the Advanced Medical Research Center, Yokohama City University (JPMXP0618217493, JPMXP0622717006, and JPMXP0723833149 to K.A.), and ONO Medical Research Foundation (to K.A.). We thank Dr. Kikuchi for generating *Lyz2*Cre/+.*Egr2*fl/fl mice in the early stage of this project, Ms. Yokoi for secretarial assistance.

## Author contributions

A.I performed the majority of experiments, with help from J.M., N.S., and K.A., and prepared figures. A.N. and T.T. performed RNA-seq analysis. Minoru, T. proofread the paper. S.S and T.S. performed scRNA-seq analysis and processed the data. T.K., T.O., and K.F. provided Egr2-flox mice and proofread the paper. Masato, T. supervised the project and proofread the paper. KA designed the study and wrote the paper.

## Competing interests

S.S. reports advisory role for ImmunoGeneTeqs, Inc; stock for ImmunoGeneTeqs, Inc. All other authors declare no competing interests.
