## [Peer Review File · Communications Biology]

Reviewers' comments:

Reviewer #1 (Remarks to the Author):

#Manuscript ID: COMMSBIO-23-3693-T

#Title: Egr2 drives the differentiation of Ly6Chi monocytes into fibrosis-promoting macrophages in non-alcoholic steatohepatitis

Ly6Chi monocytes are well-known as proinflammatory monocytes. In this study, the author described that transcription factor Egr2 can play a role in liver fibrosis by regulating the differentiation of Ly6Chi monocytes into hepatic lipid-associated macrophages (hLAMs). Using advanced analysis tools including scRNA-seq, bulk RNA-seq, and flow cytometry, the author discovered specific cell populations in which EGR2 actually acts. As cell populations associated with EGR2, TIM4-monocyte-derived macrophages were separated into cluster 6 aligned with the gene expression pattern of hLAMs and cluster 3 having transcriptional similarities with the moKCs. EGR2 regulated the proportions of hLAMs and moKCs in the monocyte-derived macrophage pool and drove the differentiation of monocytes into hLAMs. Overall, many experiments were performed, and a lot of effort was made to characterize the EGR2-related cell population. However, the author needs to explain conflicting parts compared to previous reports in diet-induced mouse models and cell types. Also, additional experiments are needed to strengthen this study.

Comment 1.

In lines 92-98, "To analyze the relationship between the development of liver fibrosis and monocyte derived macrophages, we used a mouse model of choline-deficient amino acid-defined high fat diet (CDA-HFD)-induced NASH16 that rapidly progresses to liver fibrosis. First, we analyzed the progression of NAFLD in CDA-HFD-fed mice. Microscopically, feeding of CDA-HFD promoted hepatocellular steatosis without any evidence of apparent fibrosis up to 7 weeks (Figure 1A). Compared with HFD that takes 6 months for liver fibrosis to develop¹⁷, CDA-HFD rapidly induced liver fibrosis in 12 weeks (Figures 1A-1C)."

The author described that CDA-HFD-fed mice didn't exhibit apparent fibrosis up to 7 weeks. However, it has been reported that mice fed CDA-HFD for 6 weeks developed liver fibrosis in C57BL/6J mice as confirmed by H&E and Masson's trichrome staining in reference [16] cited in this manuscript. Therefore, the author needs to explain why different results were obtained under the same experimental conditions (the same in mouse strain and diet used).

Additionally, as shown Figure 1A-C and 2E, CDA-HFD feeding for 7 weeks increased fibrosis markers and there seems to appear positive areas of Picro-Sirius Red staining. Since this study aimed to reveal the function of Egr2 at the time point for transition from NASH to fibrosis, determining the time point of fibrosis is a very important issue in this study. Therefore, histological samples need to be analyzed by an experienced pathologist.

Comment 2.

As shown in Figure 1G, EGR2 expression was markedly increased in liver macrophages of mice fed CDA-HFD for 7 weeks. The author crossed Lyz2-Cre mice with Egr2^{fl/fl} mice and confirmed depletion of Egr2 in monocytes and macrophages. However, the frequencies of monocytes, macrophages (KCs) and monocyte-derived macrophages did not differ between the two genotypes of mice at both 7 and 12 weeks. The author should explain and discuss why EGR2 deletion had no effects at this critical stage when EGR2 expression levels are high and fibrosis progression begins.

The targeted *Lyz2* gene is found in myeloid cell lineage including myeloblasts, monocytes, macrophages. CDA-HFD feeding can also occur lipid accumulation and inflammation in adipose tissues expected being recruited monocytes and macrophages. Are there changes in cell populations and signaling of monocytes and macrophages in adipose tissues at 7 and 12 weeks of CDA-HFD?

Comment 3.

In lines 136-140, "The absence of EGR2 did not change the degree of inflammation or steatosis (Figures 2C, 2D, and 2E and Figures S2C and S2D). These findings indicate that EGR2 in monocytes and monocyte-derived macrophages is not associated with the initiation of inflammation or the development of steatosis in the early stage of NAFLD."

As shown in Figures 2C and 2D and Figures S2C and S2D, there were no changes in ALT levels and TNF mRNA levels in the two genotypes of mice. However, the degree of steatosis and inflammation cannot be simply evaluated based on plasma ALT levels and TNF mRNA levels alone. Therefore, the statement that EGR2 did not affect the degree of inflammation or steatosis is overinterpreted. There are many other markers associated with steatosis and inflammation. The author needs to further investigate hepatic TG levels and markers associated with steatosis and inflammation.

Comment 4.

As shown in Figure 1G, EGR2 expression was markedly increased in liver macrophages of mice fed CDA-HFD for 7 weeks. The author crossed *Lyz2-Cre* mice with *Egr2^{fl/fl}* mice and confirmed depletion of *Egr2* in monocytes and macrophages. However, the frequencies of monocytes, macrophages (KCs) and monocyte-derived macrophages did not differ between the two genotypes of mice at both 7 and 12 weeks.

Comment 5.

In lines 150-155, "To assess the differences between monocyte-derived macrophages in NASH-induced *Egr2^{fl/fl}* mice and *Lyz2^{Cre/+}.Egr2^{fl/fl}* mice, we performed bulk RNA-seq analysis of CD11b+Ly6CloF4/80+ monocyte-derived macrophages sorted from the liver of *Lyz2^{Cre/+}.Egr2^{fl/fl}* mice and *Egr2^{fl/fl}* littermates. Unbiased clustering confirmed that the biological replicates from each group adopted a distinct transcriptional signature (Figure 3A)."

The author needs to explain the reason that can be referred as "NASH-induced *Egr2^{fl/fl}* mice" and their characteristics. In figure 3, is the CDA-HFD feeding period for mice 7 or 12 weeks? Is the CDA-HFD feeding for 7 weeks a condition for NASH development?

Comment 6.

In lines 165-167, "As the upregulation of KC markers in *Egr2*-deficient *Ly6CloF4/80+* macrophages cannot be explained by the increase of TIM4+ resident KCs at 7weeks (Figure 2B, also see scRNA-seq data discussed later in Figure 4A),"

The author needs to clarify this sentence. The author described that resident TIM4+ KCs were almost completely replaced by TIM4- monocyte-derived macrophages by week 7 in Figure 2B. If so, do TIM4- monocyte-derived macrophages have the same characteristics as *Ly6CloF4/80+* macrophages?

It is reported that TIM4+ KCs play a critical role in liver fibrosis. The author needs to discuss the extent of influence of TIM4-monocyte-derived macrophages on the progression of liver fibrosis in NASH.

Comment 7.

The author investigated the effects of long-chain saturated fatty acids as the environmental factors that trigger *Egr2* expression in the infiltrating monocytes. Are there changes in the pool of long-chain saturated and unsaturated fatty acids under conditions in which *Egr2* expression is stimulated by infiltrated monocytes in NASH?

Reviewer #2 (Remarks to the Author):

The manuscript “Egr2 drives the differentiation of Ly6Chi monocytes into fibrosis-promoting macrophages in non-alcoholic steatohepatitis” by Iwata et al. shows that the transcription factor EGR2 is involved in the differentiation of monocytes into hepatic lipid associated macrophages during metabolic dysfunction-associated steatohepatitis (MASH). In line with this, EGFR2 deficiency in vivo promotes the expression of Kupffer cells markers and correlates with decreased matrix deposition in the liver. The paper is well-written and the data are compelling. I have a few comments:

- The new nomenclature should be used throughout the paper, example metabolic dysfunction-associated steatohepatitis (MASH) instead of NASH.
- What is the mechanism by which EGFR2 induces the differentiation of monocytes into hepatic lipid associated macrophages?
- Figure 1D: immunofluorescence might be helpful to confirm the flow cytometry data.
- In vitro experiments showing the effect of EDR2+ and EGFR2- sub-populations on the activation of hepatic stellate cells are needed.

Reviewer #3 (Remarks to the Author):

In this manuscript, the authors investigated the role of EGR2 in determining the property of monocyte / macrophage population in the liver during NASH development in mice. The authors performed experiments with new technologies with a novel concept. The conclusion made by the authors has impact on the field; however, the reviewer would like to suggest several experiments to improve the quality of the paper and strengthen the conclusion by the authors.

1. The authors concluded that EGR2 is increased in experimental NASH based on their findings obtained from experiments with CDA-HFD. Is it known whether EGR2 levels are affected by other types of NASH diets, such as AMLN (GAN) diet, western diet, high-fat high-cholesterol diet, or high-fat high-fructose high-palmitate high-cholesterol diet, etc? The authors might look for the public RNA-seq data.

2. In Fig. 2, the authors showed that EGR2 KO mice had less fibrosis, but their data lack the mechanistic evidence demonstrating that profibrotic ability of macrophages are dependent on the expression of EGR2. Although the authors showed that TNFa levels were lower in Figure 6I, the reduced expression of TNFa and IL6 does not necessarily mean that these macrophages are less profibrotic. The authors could examine whether the conditioned medium from CD11c-high Mincle-high cells induced by LPS increases fibrotic gene expression in hepatic stellate cells as compared with the conditioned medium from CD11c-low Mincle-low cells induced by LPS. Also, the authors could examine whether the coculture with macrophages obtained from EGR2KO mice does not induce fibrotic gene upregulation in hepatic stellate cells as compared with the coculture with macrophages obtained from WT mice.

3. In figure 2G, the decrease of PSR is not obvious, and hydroxyproline level reduction seems marginal (about 15% reduction). So, it is advised that the authors perform qPCR analysis of other fibrotic genes,

such as TGF-beta, ACTA2, MMPs. Also, it is advised that the authors perform western blot analysis to show the decreased expression of fibrogenic markers, such as alpha-SMA.

4. In Figure 2E, the authors concluded that EGR2 deletion failed to change the degree of steatosis; however, it is difficult to determine whether lipid contents were indeed not altered with these HE images presented by the authors. Please show another evidence for lipid accumulation in the liver (oil red O staining, triglyceride measurement, etc.).

Point-by-point responses to the reviewers' comments

Reviewer #1 (Remarks to the Author):

Ly6Chi monocytes are well-known as proinflammatory monocytes. In this study, the author described that transcription factor Egr2 can play a role in liver fibrosis by regulating the differentiation of Ly6Chi monocytes into hepatic lipid-associated macrophages (hLAMs). Using advanced analysis tools including scRNA-seq, bulk RNA-seq, and flow cytometry, the author discovered specific cell populations in which EGR2 actually acts. As cell populations associated with EGR2, TIM4-monocyte-derived macrophages were separated into cluster 6 aligned with the gene expression pattern of hLAMs and cluster 3 having transcriptional similarities with the moKCs. EGR2 regulated the proportions of hLAMs and moKCs in the monocyte-derived macrophage pool and drove the differentiation of monocytes into hLAMs. Overall, many experiments were performed, and a lot of effort was made to characterize the EGR2-related cell population. However, the author needs to explain conflicting parts compared to previous reports in diet-induced mouse models and cell types. Also, additional experiments are needed to strengthen this study.

Response to the reviewer's comments

We thank the reviewer for acknowledging the value of our study and for the insightful comments that have helped improve our manuscript.

Reviewer's comment #1-1

In lines 92-98, "To analyze the relationship between the development of liver fibrosis and monocyte derived macrophages, we used a mouse model of choline-deficient amino acid-defined high fat diet (CDA-HFD)-induced NASH16 that rapidly progresses to liver fibrosis. First, we analyzed the progression of NAFLD in CDA-HFD-fed mice.

Microscopically, feeding of CDA-HFD promoted hepatocellular steatosis without any evidence of apparent fibrosis up to 7 weeks (Figure 1A). Compared with HFD that takes 6 months for liver fibrosis to develop¹⁷, CDA-HFD rapidly induced liver fibrosis in 12 weeks (Figures 1A-1C)."

The author described that CDA-HFD-fed mice didn't exhibit apparent fibrosis up to 7 weeks. However, it has been reported that mice fed CDA-HFD for 6 weeks developed liver fibrosis in C57BL/6J mice as confirmed by H&E and Masson's trichrome staining in reference [16] cited in this manuscript. Therefore, the author needs to explain why different results were obtained under the same experimental conditions (the same in mouse strain and diet used). Additionally, as shown Figure 1A-C and 2E, CDA-HFD feeding for 7 weeks increased fibrosis markers and there seems to appear positive areas of Picro-Sirius Red staining. Since this study aimed to reveal the function of Egr2 at the time point for transition from NASH to fibrosis, determining the time point of fibrosis is a very important issue in this study. Therefore, histological samples need to be analyzed by an experienced pathologist.

Response to the reviewer's comment #1-1

Matsumoto's group showed the presence of fibrotic lesion at week 6 by Masson's trichrome staining in CDA-HFD-induced NASH liver (Matsumoto, 2013 *Int J Exp Pathol*). We also found the upregulation of fibrosis-associated genes such as *Timp1* and *Colla1*, accompanied by an increase of hydroxyproline at week 7 (**Figure 1B and 1C**). As mentioned by the reviewer, these results suggested that fibrosis developed already at week 7 although collagen deposition was not obvious in terms of PSR staining compared with much more aggressive fibrosis at week 12 (**Figure 1A, right column**). Therefore, we modified our statement in the revised manuscript, lines 72-76, to clarify that CDA-HFD develops 1) hepatocellular steatosis in 7 weeks, and 2) detectable level of fibrosis as early as week 7 that rapidly progresses between week 7 and 12. We thank

the reviewer for pointing out the discrepancy between published data and our evaluation.

Reviewer's comment #1-2

As shown in Figure 1G, EGR2 expression was markedly increased in liver macrophages of mice fed CDA-HFD for 7 weeks. The author crossed Lyz2-Cre mice with Egr2^{fl/fl} mice and confirmed depletion of Egr2 in monocytes and macrophages. However, the frequencies of monocytes, macrophages (KCs) and monocyte-derived macrophages did not differ between the two genotypes of mice at both 7 and 12 weeks. The author should explain and discuss why EGR2 deletion had no effects at this critical stage when EGR2 expression levels are high and fibrosis progression begins.

Response to the reviewer's comment #1-2

As shown in the original manuscript, the proportion of monocytes and macrophages is similar between WT and Egr2 KO mice (**Figure 2B** of the original and revised manuscript). This result indicates that Egr2 is not required for the development of bulk monocyte-derived macrophages that are defined on the basis of F4/80 expression. In this study to analyze the character of monocyte-derived macrophages of WT mice and Egr2 KO mice, we found that the percentage of hLAMs was decreased, whereas percentage of moKCs was increased, in F4/80⁺ bulk monocyte-derived macrophage pool, in Egr2 KO mice (**Figure 6D and 6E** of the revised manuscript). To clarify the role of Egr2 in the development of monocyte-derived macrophages, we stated in the Result section of revised manuscript, lines 204-206, that Egr2 is controlling the proportion of hLAM and moKC subpopulations within the monocyte-derived macrophage pool. We also modified Figure 5A to clarify that F4/80⁺ monocyte-derived macrophages comprise hLAM and moKC subpopulations (Figure 5A of the revised manuscript).

Reviewer's comment #1-3

The targeted *Lyz2* gene is found in myeloid cell lineage including myeloblasts, monocytes, macrophages. CDA-HFD feeding can also occur lipid accumulation and inflammation in adipose tissues expected being recruited monocytes and macrophages. Are there changes in cell populations and signaling of monocytes and macrophages in adipose tissues at 7 and 12 weeks of CDA-HFD?

Response to the reviewer's comment #1-3

In response to the reviewer's request, we evaluated the proportion of monocytes and macrophages in adipose tissue of CDA-HFD-fed mice. Although HFD is known to induce extrahepatic adipose tissue inflammation in mice (Jaitin 2019 *Cell*, Magalhaes 2021 *Nat Commun*), our analyses showed that CDA-HFD develops NASH without inducing obvious inflammation in extrahepatic adipose tissues (**Figure for reviewer 1**).

Figure for reviewer 1. CDA-HFD does not induce extrahepatic adipose tissue inflammation. Monocytes and macrophages in epididymal adipose tissue of mice fed CDA-HFD were analyzed by flowcytometry. Ly6C^{hi}F4/80⁺ monocyte accumulation nor Tim4⁺Ly6C^{lo}F4/80⁺ monocyte-derived macrophage expansion was not observed up to 14 weeks after CDA-HFD administration. n = 3-4 mice/ time point. n.s., not significant, one-way ANOVA.

Reviewer's comment #1-4

In lines 136-140, "The absence of EGR2 did not change the degree of inflammation or steatosis (Figures 2C, 2D, and 2E and Figures S2C and S2D). These findings indicate that EGR2 in monocytes and monocyte-derived macrophages is not associated with the initiation of inflammation or the development of steatosis in the early stage of NAFLD." As shown in Figures 2C and 2D and Figures S2C and S2D, there were no changes in ALT levels and TNF mRNA levels in the two genotypes of mice. However, the degree of steatosis and inflammation cannot be simply evaluated based on plasma ALT levels and TNF mRNA levels alone. Therefore, the statement that EGR2 did not affect the degree of inflammation or steatosis is overinterpreted. There are many other markers associated with steatosis and inflammation. The author needs to further investigate hepatic TG levels and markers associated with steatosis and inflammation.

Response to the reviewer's comment #1-4

According to the reviewer's suggestion, we quantitated additional markers for inflammation such as serum AST and hepatic *I11b* mRNA level. We also evaluated the degree of steatosis on the basis of hepatic TG levels. These analyses showed that the degree of NASH-associated inflammation and steatosis was similar between two genotypes (WT and EGR2-KO) of mice, supporting our statement that EGR2 is not associated with steatosis and inflammation. We added these results as **Figure 2C (AST), 2D (I11b) and 2E (TG)** of the revised manuscript. We thank the reviewer for the constructive comment.

Reviewer's comment #1-5

As shown in Figure 1G, EGR2 expression was markedly increased in liver macrophages of mice fed CDA-HFD for 7 weeks. The author crossed Lyz2-Cre mice with Egr2^{fl/fl} mice and confirmed depletion of Egr2 in monocytes and macrophages. However, the

frequencies of monocytes, macrophages (KCs) and monocyte-derived macrophages did not differ between the two genotypes of mice at both 7 and 12 weeks.

Response to the reviewer's comment #1-5

I apologize for confusing the reviewer by not clearly explaining the role of EGR2 in monocyte and macrophage differentiation in NASH. As we answered in reviewer's comment #1-2, *Egr2* is dispensable for the differentiation of bulk monocyte-derived macrophages that are defined on the basis of F4/80 expression but is controlling the proportion of hLAM and moKC subpopulations within the F4/80⁺ monocyte-derived macrophage pool (**Figure 5A** of the revised manuscript). Please also see our response to reviewer's comment #1-2.

Reviewer's comment #1-6

*In lines 150-155, "To assess the differences between monocyte-derived macrophages in NASH-induced *Egr2*^{fl/fl} mice and *Lyz2*^{Cre/+}.*Egr2*^{fl/fl} mice, we performed bulk RNA-seq analysis of CD11b⁺Ly6CloF4/80⁺ monocyte-derived macrophages sorted from the liver of *Lyz2*^{Cre/+}.*Egr2*^{fl/fl} mice and *Egr2*^{fl/fl} littermates. Unbiased clustering confirmed that the biological replicates from each group adopted a distinct transcriptional signature (Figure 3A). "The author needs to explain the reason that can be referred as "NASH-induced *Egr2*^{fl/fl} mice" and their characteristics. In figure 3, is the CDA-HFD feeding period for mice 7 or 12 weeks? Is the CDA-HFD feeding for 7 weeks a condition for NASH development?"*

Response to the reviewer's comment #1-6

The reviewer requests that we explain why we defined mice fed CDA-HFD for 7 weeks as NASH-induced *Egr2*^{fl/fl} mice and *Lyz2*^{Cre/+}.*Egr2*^{fl/fl} mice. As we explained in Response to the reviewer #1-1, on the basis of qPCR analysis and hydroxyproline assay,

we modified our statement in the revised manuscript (lines 72-76) to clarify that CDA-HFD develops hepatocellular steatosis and liver fibrosis (NASH) as early as week 7 that further progresses in the following weeks.

Reviewer's comment #1-7-1

In lines 165-167, "As the upregulation of KC markers in Egr2-deficient Ly6CloF4/80+ macrophages cannot be explained by the increase of TIM4+ resident KCs at 7weeks (Figure 2B, also see scRNA-seq data discussed later in Figure 4A)," The author needs to clarify this sentence. The author described that resident TIM4+ KCs were almost completely replaced by TIM4- monocyte-derived macrophages by week 7 in Figure 2B. If so, do TIM4- monocyte-derived macrophages have the same characteristics as Ly6CloF4/80+ macrophages?

Response to the reviewer's comment #1-7-1

We thank the reviewer for raising an important question. We found enhanced KC marker expression in bulk F4/80⁺ monocyte-derived macrophages in Egr2-deficient mice (**Figure 3E**). From this result, we initially presumed that the proportion of KCs was increased in F4/80⁺ macrophage fraction in Egr2-deficient NASH mice. Contrary to our hypothesis, the scRNA-seq analysis showed that Tim4⁺ resident KCs were barely observed at week 7 both in WT and Egr2-deficient mice (**Figure 4A, right four panels, clusters 3 and 14**). Instead, we found that Egr2-deficiency caused expansion of moKCs that are distinct from either resident KCs or hLAMs. In order not to confuse readers, we carefully explained that the enhanced expression of KC markers cannot be attributed to the accumulation of KCs within bulk F4/80⁺ macrophage pool in Egr2-deficient mice in the RESULT section, lines 186-193, of revised manuscript.

Reviewer's comment #1-7-2

It is reported that TIM4⁺ KCs play a critical role in liver fibrosis. The author needs to discuss the extent of influence of TIM4-monocyte-derived macrophages on the progression of liver fibrosis in NASH.

Response to the reviewer's comment #1-7-2

The reviewer requests that that we discuss the contribution of KCs and monocyte-derived macrophages on the progression of liver fibrosis in NASH. According to the reviewer's request, we discussed roles played by Tim4⁺ KCs and Tim4⁻ monocyte-derived macrophages in the development of NAFLD in the Discussion section of the revised manuscript, lines 315-319 with appropriate citations.

Reviewer's comment #1-8

The author investigated the effects of long-chain saturated fatty acids as the environmental factors that trigger Egr2 expression in the infiltrating monocytes. Are there changes in the pool of long-chain saturated and unsaturated fatty acids under conditions in which Egr2 expression is stimulated by infiltrated monocytes in NASH?

Response to the reviewer's comment #1-8

This reviewer is asking if the pool of long-chain fatty acids changes in NASH. As mentioned in result section and discussion section of the original manuscript, Chiappini's group performed comprehensive lipidomic analysis on human liver biopsies of healthy and NASH patients. Their analysis identified 32 lipids including long-chain saturated fatty acids; C16:0 and C14:0, and unsaturated fatty acids; C16:1n-7, C16:1n-9, C18:1n-7, and C18:2n-6, discriminating NASH with 100% accuracy (Chiappini, 2016 *Sci Rep*). Accumulation of saturated long-chain fatty acids is reported also in a mouse model of NASH (Spann, 2012 *Cell*). We added another citation (#31) to the reference list of the revised manuscript to underscore the role that long-chain saturated fatty acids

may play over infiltrating monocytes.

Reviewer #2 (Remarks to the Author):

The manuscript “Egr2 drives the differentiation of Ly6Chi monocytes into fibrosis-promoting macrophages in non-alcoholic steatohepatitis” by Iwata et al. shows that the transcription factor EGR2 is involved in the differentiation of monocytes into hepatic lipid associated macrophages during metabolic dysfunction-associated steatohepatitis (MASH). In line with this, EGFR2 deficiency in vivo promotes the expression of Kupffer cells markers and correlates with decreased matrix deposition in the liver. The paper is well-written and the data are compelling. I have a few comments:

Response to the reviewer’s comments

We thank the reviewer for acknowledging the value of our study and for the insightful comments that have helped improve our manuscript.

Reviewer’s comment #2-1

- The new nomenclature should be used throughout the paper, example metabolic dysfunction-associated steatohepatitis (MASH) instead of NASH.

Response to the reviewer’s comment #2-1

As pointed by the reviewer, the new nomenclature for fatty liver diseases, metabolic dysfunction-associated steatotic liver diseases, was introduced in 2023 (Rinella, 2023 *J. Hepatol*). Accordingly, we replaced NAFLD/NASH with MASLD/MASH throughout the paper.

Reviewer’s comment #2-2

- What is the mechanism by which EGFR2 induces the differentiation of monocytes into

hepatic lipid associated macrophages?

Response to the reviewer's comment #2-2

We thank the reviewer for raising an important question that was not totally answered in this study. We showed that the expression of several markers for SAM/hLAMs are repressed in monocyte-derived macrophages of Egr2-deficient mice (**Figures 3C-3E**). On the other hand, expression of transcription factors known to control liver macrophage differentiation, e.g. *Atf3*, *Pparg*, *Nr1h3*, and *Rbpj*, was not largely affected by the absence of Egr2 (**Supplementary Figure 3C** of the revised manuscript). This finding suggests that Egr2 is not controlling the expression of master regulators of liver macrophages. From these results, we speculate that Egr2 either directly or indirectly control the acquisition of SAM/hLAM characteristics (e.g. CD11c) during maturation from monocytes to macrophages without impeding the differentiation of monocyte-derived macrophages. In the revised manuscript, we stated that we could not determine exactly how EGR2 control the character of monocyte-derived macrophages and discussed the possible mechanism by which EGR2 controls the differentiation of monocytes into hLAMs, lines 330-335. We also added TF expressions as Supplementary Figure 3C of the revised manuscript.

Reviewer's comment #2-3

- *Figure 1D: immunofluorescence might be helpful to confirm the flow cytometry data.*

Response to the reviewer's comment #2-3

We thank the reviewer for the constructive comment. In response to the reviewer's advice, we analyzed F4/80 and Tim4 expression in naïve and NASH liver by immunohistochemistry. This analysis clearly demonstrated that Tim4⁺ cells drastically decrease in macrophages in NASH liver. Also, this analysis showed that aggregates of

F4/80⁺ macrophages are forming crown-like structures that may be surrounding dead hepatocytes in NASH liver. We added these findings as **Figure 1E** along with sentences, lines 81-85, in the Result section of the revised manuscript.

Reviewer's comment #2-4

- In vitro experiments showing the effect of EDNR2⁺ and EGFR2⁻ sub-populations on the activation of hepatic stellate cells are needed.

Response to the reviewer's comment #2-4

We agree with the reviewer's comment. In response to the reviewer's request, we sorted liver F4/80⁺CD11c^{hi}Mincle^{hi} macrophages (hLAMs) and F4/80⁺CD11c^{lo}Mincle^{lo} macrophages (moKCs) from NASH-induced WT mice and stimulated them *in vitro* with 0.1 µg/mL LPS for 18 h. Then, hepatic stellate cells (HSCs) that were purified by density gradient centrifugation were cultured for 18 h in the presence of conditioned media (CM) collected from LPS-stimulated sorted macrophages. Fibrotic marker genes in those HSCs were measured by qPCR. In four to five out of six mice tested, *Timp1*, *Colla1* and *Acta2* levels were decreased in HSCs cultured in CD11c^{lo}Mincle^{lo}-macrophage-CM compared to those cultured in CD11c^{hi}Mincle^{hi}-macrophage-CM (p=0.45, 0.078 and 0.14, respectively, **Figure for reviewer 2**). From these results, we speculated that CD11c^{lo}Mincle^{lo} macrophages are less fibrotic than CD11c^{hi}Mincle^{hi} macrophages. However, we prefer not to include this data in the revised manuscript, because to reach a solid conclusion, it necessitates a significantly extended timeframe and the use of more animals. We thank the reviewer for understanding our situation and helpful suggestions to address the issue. Please also see our response to the reviewer #3-2.

Figure for reviewer 2. *In vitro* HSC-culture assay. Primary hepatic stellate cells were cultured in the presence of LPS-stimulated CD11c^{hi}Mincle^{hi} macrophages or CD11c^{lo}Mincle^{lo} macrophages for 18 h. HSC activation markers (*Timp1*, *Col1a1*, or *Acta2*) were quantitated by PCR. Each symbol represents an individual animal. Paired t-test.

Reviewer #3 (Remarks to the Author):

In this manuscript, the authors investigated the role of EGR2 in determining the property of monocyte / macrophage population in the liver during NASH development in mice. The authors performed experiments with new technologies with a novel concept. The conclusion made by the authors has impact on the field; however, the reviewer would like to suggest several experiments to improve the quality of the paper and strengthen the conclusion by the authors.

Response to the reviewer’s comments

We thank the reviewer for acknowledging the value of our study and for the insightful comments that have helped improve our manuscript.

Reviewer’s comment #3-1

The authors concluded that EGR2 is increased in experimental NASH based on their

findings obtained from experiments with CDA-HFD. Is it known whether EGR2 levels are affected by other types of NASH diets, such as AMLN (GAN) diet, western diet, high-fat high-cholesterol diet, or high-fat high-fructose high-palmitate high-cholesterol diet, etc? The authors might look for the public RNA-seq data.

Response to the reviewer's comment #3-1

We thank this reviewer for the constructive advice. Egr2 is included in the list of genes that are upregulated in HFD- and AMLN-induced NASH models in mice (Daemen, 2021 *Cell Rep*, Seidman, 2020 *Immunity*). Ramachandran's group reported that EGR2 level is enhanced also in human NASH patient (Ramachandran, 2019 *Nature*). We introduced these findings in Discussion section of the revised manuscript, with appropriate citations (#9, 11, 13), lines 360-364.

Reviewer's comment #3-2

In Fig. 2, the authors showed that EGR2 KO mice had less fibrosis, but their data lack the mechanistic evidence demonstrating that profibrotic ability of macrophages are dependent on the expression of EGR2. Although the authors showed that TNFa levels were lower in Figure 6I, the reduced expression of TNFa and IL6 does not necessarily mean that these macrophages are less profibrotic. The authors could examine whether the conditioned medium from CD11c-high Mincle-high cells induced by LPS increases fibrotic gene expression in hepatic stellate cells as compared with the conditioned medium from CD11c-low Mincle-low cells induced by LPS. Also, the authors could examine whether the coculture with macrophages obtained from EGR2KO mice does not induce fibrotic gene upregulation in hepatic stellate cells as compared with the coculture with macrophages obtained from WT mice.

Response to the reviewer's comment #3-2

We thank the reviewer for helpful suggestions. In response to the reviewer's request, we sorted liver F4/80⁺CD11c^{hi}Mincle^{hi} macrophages (hLAMs) and F4/80⁺CD11c^{lo}Mincle^{lo} macrophages (moKCs) from NASH-induced WT mice and stimulated them *in vitro* with 0.1 µg/mL LPS. Then, hepatic stellate cells (HSCs) that were purified by density gradient centrifugation were cultured for 18 h in the presence of conditioned media (CM) collected from LPS-stimulated sorted macrophages. Fibrotic marker genes in those HSCs were measured by qPCR. In four to five out of six mice tested, *Timp1*, *Colla1*, and *Acta2* levels were decreased in HSCs cultured in CD11c^{lo}Mincle^{lo}-macrophage-CM compared with those cultured in CD11c^{hi}Mincle^{hi}-macrophage-CM (p = 0.45, 0.078 and 0.14, respectively, **Figure for reviewer 2**). It appears that CD11c^{lo}Mincle^{lo} macrophages are less fibrotic than CD11c^{hi}Mincle^{hi} macrophages, however, we would prefer not to include this data in the revised manuscript because to reach a solid conclusion, it necessitates a significantly extended timeframe and the use of more animals. We thank the reviewer for understanding our situation. Please also see our response to the reviewer #2-4.

Reviewer's comment #3-3

In figure 2G, the decrease of PSR is not obvious, and hydroxyproline level reduction seems marginal (about 15% reduction). So, it is advised that the authors perform qPCR analysis of other fibrotic genes, such as TGF-beta, ACTA2, MMPs. Also, it is advised that the authors perform western blot analysis to show the decreased expression of fibrogenic markers, such as alpha-SMA.

Response to the reviewer's comment #3-3

According to the reviewer's advice, we quantitated a broader range of fibrosis-associated genes other than *Timp1* and *Colla1*. This analysis showed that *Tgfb1* and *Acta2* expressions were also reduced in Egr2 KO mice. We further performed western

blotting analysis of liver tissue of WT and Egr2-KO mice. The amount of liver alpha-SMA appeared smaller in *Egr2*-KO mice than in WT mice, although the difference did not reach a statistical significance ($p = 0.211$) (**Figure for reviewer 3**). We added qPCR data as **Figure 2I** of revised manuscript. We thank the reviewer again for helpful advice to strengthen the importance of Egr2 on the progression of liver fibrosis.

Figure for reviewer 3. Western blotting showing the reduction of α -SMA in EGR2-KO mouse liver (left). Signals were quantitated with Fusion Solo S analyzer (Vilber, France). $n=4$ mice/genotype. Each lane or symbol represents an individual animal (right).

Reviewer's comment #3-4

In Figure 2E, the authors concluded that EGR2 deletion failed to change the degree of steatosis; however, it is difficult to determine whether lipid contents were indeed not altered with these HE images presented by the authors. Please show another evidence for lipid accumulation in the liver (oil red O staining, triglyceride measurement, etc.).

Response to the reviewer's comment #3-4

In response to the reviewer's request, we quantitated liver TG level as another indicator of lipid accumulation. This analysis showed that liver TG levels are similar between WT and EGR2-KO mice. We added this finding as **Figure 2E** of the revised manuscript.

Please also see our response to Reviewer #1-4.

Reviewers' comments:

Reviewer #1 (Remarks to the Author):

#Manuscript ID: COMMSBIO-23-3693-A

#Title: Egr2 drives the differentiation of Ly6Chi monocytes into fibrosis-promoting macrophages in metabolic dysfunction-associated steatohepatitis

This is a very interesting and deep study. The author has tried to address the comments. Overall, scientific content and readability have been strengthened. However, there are still some fundamental questions that need to be resolved.

Comment 1.

In lines 74-76, "liver fibrosis was detectable at week 7 and further progressed within the next 5 weeks by feeding CDA-HFD (Figures 1A-1C)."

In lines 117-121, "The absence of Egr2 did not change the degree of inflammation or steatosis (Figures 2C, 2D, and 2E and Figures S2C and S2D). These findings indicate that Egr2 in monocytes and monocyte-derived macrophages is not associated with the initiation of inflammation or the development of steatosis in the early stage of MASLD."

The author described liver fibrosis was detectable at week 7 of feeding CDA-HFD. The author needs to explain whether this condition can be called the initiation of inflammation or the early stage of NAFLD and to confirm whether steatosis and inflammation were actively during this period.

Because this study can be explained through the function of immune cells that form the basis of the inflammatory response. Additionally, an increase in immune cells in the liver may affect the research results.

Comment 2.

In lines 148-151, "these findings suggest that Egr2 is required for the full equipment of SAM phenotype by monocyte derived macrophages, which may promote the transition from simple steatosis to fibrosis over the course of MASLD."

The biggest question is whether activation and inflammatory responses of immune cells can be excluded from the process from steatosis to fibrosis.

Additionally, the author demonstrated that Egr2 deficiency led to moKC expansion, and enhanced KC marker expression in bulk F4/80+ monocyte-derived macrophages. However, the author explained that this does not imply KCs accumulation within bulk F4/80+ macrophage pool in Egr2-deficient mice. There must have been changes between the various cell populations, but it's unclear and confusing. In other word, it is not clear whether monocytes from other sources were introduced into the liver and transformed into macrophages, or whether resident monocytes differentiated into macrophages, and changes within the cell population due to cell differentiation also need to be explained.

Minor comment.

In line 82, "Tim4 expression was reduced in macrophages in NASH liver (Figure 1E)."

The author needs to modify NASH to MASH throughout the manuscript for consistency.

Reviewer #2 (Remarks to the Author):

I thank the authors for their responses. I have no further comments.

Reviewer #3 (Remarks to the Author):

The authors adequately addressed the questions raised by the reviewer.

Point-by-point responses to the reviewers' comments

Reviewer #1 (Remarks to the Author):

This is a very interesting and deep study. The author has tried to address the comments. Overall, scientific content and readability have been strengthened. However, there are still some fundamental questions that need to be resolved.

Response to the comment:

We thank the reviewer for acknowledging the value of our study. Any change in the manuscript is highlighted in yellow.

Comment 1.

In lines 74-76, "liver fibrosis was detectable at week 7 and further progressed within the next 5 weeks by feeding CDA-HFD (Figures 1A-1C)."

In lines 117-121, "The absence of Egr2 did not change the degree of inflammation or steatosis (Figures 2C, 2D, and 2E and Figures S2C and S2D). These findings indicate that Egr2 in monocytes and monocyte-derived macrophages is not associated with the initiation of inflammation or the development of steatosis in the early stage of MASLD."

The author described liver fibrosis was detectable at week 7 of feeding CDA-HFD. The author needs to explain whether this condition can be called the initiation of inflammation or the early stage of NAFLD and to confirm whether steatosis and inflammation were actively during this period. Because this study can be explained through the function of immune cells that form the basis of the inflammatory response. Additionally, an increase in immune cells in the liver may affect the research results.

Response to the Comment 1.

To be consistent, we toned down the conclusion by omitting the following sentence,

“These findings indicate that Egr2 in monocytes and monocyte-derived macrophages is not associated with the initiation of inflammation or the development of steatosis in the early stage of MASLD”, in the revised manuscript.

Comment 2-1.

In lines 148-151, “these findings suggest that Egr2 is required for the full equipment of SAM phenotype by monocyte derived macrophages, which may promote the transition from simple steatosis to fibrosis over the course of MASLD.”

The biggest question is whether activation and inflammatory responses of immune cells can be excluded from the process from steatosis to fibrosis.

Additionally, the author demonstrated that Egr2 deficiency led to moKC expansion, and enhanced KC marker expression in bulk F4/80+ monocyte-derived macrophages.

However, the author explained that this does not imply KCs accumulation within bulk F4/80+ macrophage pool in Egr2-deficient mice. There must have been changes between the various cell populations, but it's unclear and confusing.

Response to the Comment 2-1.

Although the proportion of T cells and B cells was similar between wild type and Egr2-deficient mice spleen (**Supplementary Figure 2A**), we did not evaluate the changes in various immune cell types in detail in MASH liver. To acknowledge this limitation, we stated that immune cells other than monocytes and macrophages may also contribute to the progression of liver fibrosis, lines 334-336, in the Discussion section of the revised manuscript.

Comment 2-2.

In other word, it is not clear whether monocytes from other sources were introduced into the liver and transformed into macrophages, or whether resident monocytes

differentiated into macrophages, and changes within the cell population due to cell differentiation also need to be explained.

Response to the Comment 2-2.

We thank the reviewer for raising an interesting point. As mentioned by the reviewer, our scRNA-seq analysis identified two transcriptionally distinct Ly6C^{hi} monocytes, clusters 1 and 4, in MASH liver (**Figures 4B, C, and D**). Although the RNA velocity analysis did not predict the differentiation of monocyte cluster 4 into macrophages (**Figure 5A**), we discussed that moKCs and hLAMs may arise from different monocytes in the Discussion section of the revised manuscript (lines 321-325).

Minor comment

In line 82, “Tim4 expression was reduced in macrophages in NASH liver (Figure 1E).” The author needs to modify NASH to MASH throughout the manuscript for consistency.

Response to the Minor comment.

We replaced NASH with MASH throughout the manuscript, lines 86 and 88. We thank the reviewer for careful proofreading.

Reviewer #2 (Remarks to the Author):

I thank the authors for their responses. I have no further comments.

Response to the reviewer.

We thank the reviewer for helpful comments to improve the value of our study.

Reviewer #3 (Remarks to the Author):

The authors adequately addressed the questions raised by the reviewer.

Response to the reviewer.

We thank the reviewer for helpful comments to improve the value of our study.

REVIEWERS' COMMENTS:

Reviewer #1 (Remarks to the Author):

The author constructively resolved the comments raised by the reviewer. There are no further comments and questions.

Responses to the reviewers' comments

Reviewer #1 (Remarks to the Author):

The author constructively resolved the comments raised by the reviewer. There are no further comments and questions.

Response to the comment:

We thank the reviewer for constructive comments to improve the value of our manuscript.